# BrainFCIR: Functional Context Informed Representation Learning for Intracranial Neural Signals

## Abstract

Intracranial neural recordings (e.g., stereo-ElectroEncephaloGraphy (sEEG)) have offered a unique window to measure neural signals across multiple brain regions simultaneously. Recent works have focused on developing neurofoundation models that learn generalizable representations across both subjects and tasks from such recordings. These models achieve exciting advances, yet overlook the modular functional organization of the brain, where neurons from multiple adjacent anatomical regions collectively support specific cognitive functions (e.g., the Wernicke area for speech perception). A key challenge remains how to effectively incorporate this functional contextual information into representation learning to improve both interpretability and decoding performance. To tackle this challenge, we propose a novel pre-training framework, BrainFCIR, that explicitly integrates functional context into model design via spatial-context-guided representation learning. We evaluate BrainFCIR on the publicly available sEEG speech-perception dataset. Extensive experiments show that BrainFCIR, as a unified representation learning framework for intracranial sEEG signals, significantly outperforms previous decoding methods. Overall, our work underscores the significance of functional context in developing more biologically plausible and high-performing neural decoding models. Code and checkpoints will be publicly available.

## 1 Introduction

Intracranial stereo-ElectroEncephaloGraphy (sEEG) provides a unique window into human brain function by recording neural activity directly from deep brain structures with high temporal precision. This methodology enables simultaneous monitoring across distributed brain regions, capturing dynamic interactions that underlie complex cognitive and behavioral processes. The capacity to model these high-resolution signals offers significant potential for advancing our understanding of large-scale functional networks (Wang et al., 2023; Zhang et al., 2023; Mentzelopoulos et al., 2024; Zheng et al., 2025; Chau et al., 2024), with important implications for developing next-generation neurotechnologies such as closed-loop brain-computer interfaces. In contrast to non-invasive approaches like functional Magnetic Resonance Imaging (fMRI) (Caro et al., 2023; Dong et al., 2024) or ElectroEncephaloGraphy (EEG) (Jiang et al., 2024b; Wang et al., 2024c; Jiang et al., 2024a), sEEG bypasses the signal attenuation caused by the skull and scalp, providing more direct measurements of neural dynamics with millisecond-scale resolution. Furthermore, while intracortical Micro-Electrode Array (MEA) typically samples activity from highly localized neuronal populations, sEEG offers broader coverage across multiple brain systems, albeit with sparser spatial sampling. This combination of high temporal resolution and wide spatial coverage presents both unique opportunities and distinctive computational challenges for modeling the brain's spatiotemporal organization.

The development of foundation models for intracranial sEEG has become an active area of research, driven by the goal of learning generalizable spatiotemporal representations of neural activity. This effort mirrors similar advances in modeling other neural data modalities, including intracortical MEA (Azabou et al., 2023; Ye et al., 2023; 2025), non-invasive EEG (Jiang et al., 2024b; Wang et al., 2024c; Jiang et al., 2024a), and fMRI (Caro et al., 2023; Dong et al., 2024). Contemporary approaches frequently employ large-scale transformer architectures, pre-trained in a self-supervised manner, to learn powerful representations from sEEG that demonstrate strong performance on downstream tasks

and robust cross-subject generalization (Wang et al., 2023; Zhang et al., 2023; Chau et al., 2024). A key question, however, concerns the optimal method for training these models to better capture functional connectivity, which is crucial for both neural decoding and functional groups identification.

While prior works (Wang et al., 2023; Zhang et al., 2023; Li et al., 2025) have largely used mask-based reconstruction tasks to understand the spatial-temporal organization of sEEG recordings, these models may over-rely on the intra-channel temporal patterns, leaving it unclear whether these models effectively capture functional context. PopT (Chau et al., 2024) takes the first step in effectively modeling inter-channel context during the pre-training stage. However, their approach decouples temporal and spatial modeling into two isolated stages (i.e., BrainBERT (Wang et al., 2023) for temporal modeling and PopT (Chau et al., 2024) for spatial modeling), preventing effective interaction between them – temporal-transformed embeddings cannot be refined leveraging spatial context, which ultimately limits the effectiveness of subsequent spatial modeling. As such, developing models that enable capturing precise inter-channel interaction (i.e., functional context) and studying the channel cluster estimated by such a neurofoundation model remains unexplored.

To address these two issues, we propose a functional-context-informed neurofoundation model for intracranial sEEG recordings, BrainFCIR, which models spatial-temporal relationships through functional context discrimination. To quantify the effect of functional context modeling, we estimate functional connectivity via the pre-trained model. And we perform channel clustering to identify functional groups, which enhances neural decoding via channel selection and further demonstrates its superiority in capturing inter-channel interactions.

To validate the effectiveness of our proposed framework, we evaluate BrainFCIR on the widely used Brain Treebank dataset (Appendix A). Empirically, BrainFCIR outperforms existing neurofoundation models (Zhang et al., 2023; Chau et al., 2024) for sEEG recordings and identifies channel cluster that faithfully aligns with those containing target neural activity. Besides, we further evaluate the capability of cross-subject functional group identification, demonstrating the great potential of BrainFCIR to support functionally grouping sEEG channels from unseen subjects.

To sum up, the main contributions of our work comprise:

1. **Explicit functional-context modeling:** We develop a spatiotemporal transformer model, BrainFCIR, for intracranial sEEG recordings and an associated functional context discrimination task. During the pre-training stage, we explicitly model functional context to encourage learning functional-context-informed representations.

2. **State-of-the-art (SOTA) performance:** Our model achieves SOTA performance in decoding speech perception from intracranial sEEG signals on the Brain Treebank dataset, demonstrating robust effectiveness across diverse decoding tasks.

3. **Cross-subject functional group identification:** Our model shows the potential to offer an off-the-shelf functional group identification toolkit for unseen subjects. When pre-trained within the target subject, functional groups identified by our model further enhance decoding.

## 2 RELATED WORKS

### 2.1 SPATIOTEMPORAL MODELS OF INTRACRANIAL SEEG RECORDINGS

Several prior studies have proposed spatiotemporal models for sEEG modeling, employing various strategies to incorporate spatial information. While the initial version of Brant introduced by Zhang et al. (2023) did not explicitly encode spatial relationships, later iterations (Li et al., 2025) incorporated learnable positional embeddings along the spatial axis, albeit without integrating neuroanatomical priors. Zheng et al. (2025) adopted a region-level approach, in which all channels within the pre-defined brain region were pooled, effectively collapsing spatial variability and eliminating the need for fine-grained spatial encoding. In comparison, Mentzelopoulos et al. (2024) and Chau et al. (2024) modeled space at the single-channel level by deriving token-wise spatial encodings from neuroanatomical coordinates of each channel, thereby incorporating anatomical context.

Despite these advances, to the best of our knowledge, most previous sEEG modeling studies (Zhang et al., 2023; Li et al., 2025; Zheng et al., 2025) rely on the mask-based reconstruction task to learn spatial-temporal relationships of intracranial sEEG recordings, which may over-rely on the intra-

channel temporal patterns. PopT (Chau et al., 2024) takes the first step in effectively capturing inter-channel context, despite their isolated spatial-temporal modeling strategies preventing effective interaction between spatial and temporal modeling.

## 2.2 SELF-SUPERVISED LEARNING IN BCI

Recently, the pre-trained temporal-spatial models (i.e., foundation models) have drawn significant attention across diverse neural modalities, including EEG (Jiang et al., 2024b; Wang et al., 2024b;c), fMRI (Caro et al., 2023; Dong et al., 2024), neural spike (Ye et al., 2023; Zhang et al., 2024), etc. Since these neural modalities either have lower spatial resolution compared to sEEG recordings (e.g., EEG, fMRI) or are typically implanted within a specific brain region (e.g., neural spike), they mainly leverage masked self-supervised pre-training for spatiotemporal models. However, intracranial sEEG recordings are inherently different from those recordings, providing a unique window into human brain function by recording eural activity directly from deep brain structures with high temporal & spatial resolution. This feature requires sEEG modeling methods to accurately identify channels from different functional modules, thereby enhancing our understanding of channel interactions within each functional group and improving neural decoding performance. As such, we develop a spatiotemporal transformer model paired with a functional context discrimination task, which enables the effective capture of inter-channel relationships for downstream decoding. Finally, to further quantify the effect of functional context modeling, we identify functional groups based on the estimated functional connectivity, further enhancing downstream decoding.

## 3 METHOD

To effectively capture functional context in representation learning, we develop a spatiotemporal transformer model and a novel pre-training framework that effectively guides the model to capture the functional context, thus enhancing downstream decoding. We first describe our spatial-temporal transformer model architecture. We then detail how our self-supervised pre-training procedure guides the model to learn functional-context-informed representations. Finally, we discuss our evaluation schemes.

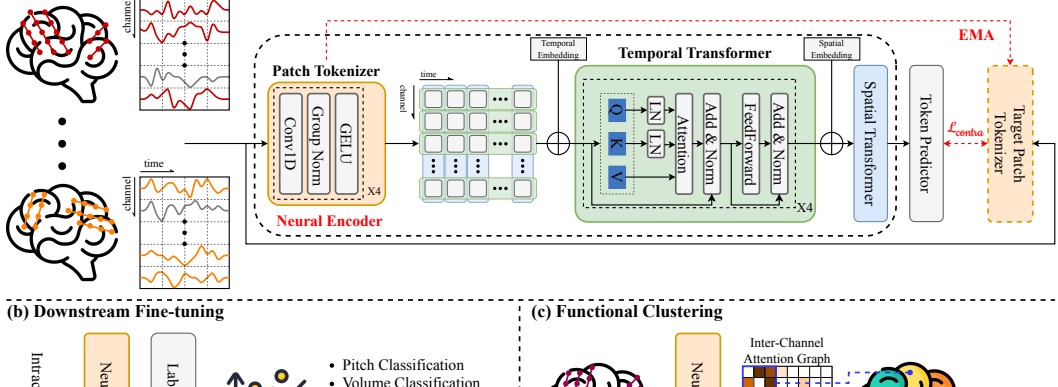

Figure 1: **Overview of BrainFCIR model. (a).** BrainFCIR is pre-trained via functional context discrimination. **(b).** Randomly initialized MLP is stacked on the pre-trained Neural Encoder to support downstream fine-tuning. **(c).** BrainFCIR supports functional connectivity estimation (even for new subjects), identifying functional groups to enhance downstream decoding.

## 3.1 MODEL ARCHITECTURE

Our model architecture and tokenization scheme are shown in Figure 1 (a). Given a multivariate time series of intracranial sEEG activity $X \in \mathbb{R}^{C \times T}$, where $C$ denotes the number of recording channels and $T$ denotes the total timestamps, we first tokenize channels as univariate signals (i.e., agnostic to space), following common practice (Zhang et al., 2023; Chau et al., 2024; Jiang et al., 2024b). We create temporal patches of each channel that are of length $L$ (e.g., 250 milliseconds), yielding $\mathcal{X}_p = \{\boldsymbol{x}_{i,j}^p \in \mathbb{R}^L | i = 1, ..., C; j = 1, ..., N\}$, where $N = \lfloor \frac{T}{L} \rfloor$, the number of patches is $|\mathcal{X}_p| = C \times N$, and $\boldsymbol{x}_{i,j}^p \in \mathbb{R}^L$ indicates the $i$-th patch of length $L$ for the $j$-th channel.

The model architecture of BrainFCIR comprises two parts: (1) Patch Tokenizer; (2) Temporal & Spatial Transformer. The Patch Tokenizer consists of a stack of convolution blocks. In the first step of tokenization, each patch $\mathcal{X}_{i,j}^p$ is passed through the Patch Tokenizer (shared across patches). In practice this tokenizer can take any form; here we choose a temporal convolution neural network (CNN) both to account for the input signal's continuous nature and because of prior domain knowledge about the importance of oscillatory features in neural activity (Jacobs & Kahana, 2010; Buzsaki & Draguhn, 2004). In each convolution block, the temporal convolution layer is stacked with group normalization (Wu & He, 2018), and Gaussian Error Linear Unit (GELU) activation (Hendrycks & Gimpel, 2016). We denote the patch embeddings from the Patch Tokenizer as

$$\mathcal{E}_p = \{\boldsymbol{e}_{i,j}^p \in \mathbb{R}^d | i = 1, ..., C; j = 1, ..., N\}, \tag{1}$$

where $d$ is the dimension of embeddings.

In order to enable the model to be aware of the temporal information of patch embeddings, we utilize the parameter-free temporal embeddings introduced by (Vaswani et al., 2017), i.e., $\mathcal{E}_t = \{\boldsymbol{e}_1^t, ..., \boldsymbol{e}_{t_{max}}^t\}$. Note that $t_{max}$ is the hyperparameter determining the maximum number of time patches and $t_{max} \geq N$. Given one arbitrary patch embedding $\boldsymbol{e}_{i,j}^p$ in Equation 1, we add the corresponding temporal embedding to it:

$$\mathcal{E} = \{\boldsymbol{e}_{i,j}^p + \boldsymbol{e}_j^t | i = 1, ..., C; j = 1, ..., N\}, \tag{2}$$

which forms the input embeddings $\mathcal{E}$ for the Temporal Transformer. Then, the embeddings will be directly fed into the Transformer encoder (Vaswani et al., 2017) to get the temporal-transformed embeddings $\mathcal{E} = \{\boldsymbol{e}_{i,j} | i = 1, ..., C; j = 1, ..., N\}$.

The functions supported by the same anatomical brain regions in different subjects are roughly similar (e.g., the superior temporal gyrus (STG) consistently participates in auditory perception), although fine-grained functional sub-organization may vary between subjects (Buzsáki, 2006). To incorporate such anatomical priors into our model, we encode standardized anatomical coordinates (e.g., LPI coordinates, MNI coordinates) using sinusoidal position encoding, generating a set of spatial embeddings $\mathcal{E}_s = \{\boldsymbol{e}_i^s \in \mathbb{R}^d | i = 1, ..., C\}$. These spatial embeddings are then added to the corresponding temporal-transformed embeddings to form the combined input $\mathcal{E} = \{\boldsymbol{e}_{i,j} + \boldsymbol{e}_i^s | i = 1, ..., C; j = 1, ..., N\}$, which is subsequently processed by the Spatial Transformer to produce the spatial-transformed embeddings $\mathcal{E}$.

To improve the stability and efficiency of transformer training, we adopt several optimizations proposed by Dehghani et al. (2023). These include applying layer normalization to queries and keys before computing dot-product attention, which prevents extreme values in attention logits and promotes more stable gradient dynamics during learning:

$$\text{Attention}(Q, K, V) = \text{softmax}(\frac{\text{LN}(Q)\text{LN}(K)^T}{\sqrt{d_{head}}})V, \tag{3}$$

where $d_{head}$ is the dimension of attention head and LN denotes layer normalization (Ba et al., 2016).

## 3.2 REPLACED FUNCTIONAL CONTEXT DISCRIMINATION

To encourage the model to effectively capture inter-channel functional context, we train BrainFCIR using a replaced functional context discrimination task (Figure 1 (a)), which differs from prior work in two ways. First, we follow JEPA (Assran et al., 2023) to encourage the model to discriminate replaced functional context in the latent token space instead of the observation space (Zhang et al., 2023; Li et al., 2025), which enhances the quality of learned representations. Second, unlike some prior

intracranial sEEG models (Wang et al., 2023; Zheng et al., 2025; Chau et al., 2024), we simultaneously train both the tokenizer and spatial-temporal Transformer to perform replaced functional context discrimination.

During pre-training, for each sEEG sample $\mathcal{X} \in \mathbb{R}^{C \times T}$, 10% of channels are randomly selected to have their activity replaced by activity from unrelated time points. To ensure balanced label distribution, we designate only 10% of unreplaced channels as positive samples during pre-training. The modified sample is directly fed into the Patch Tokenizer to get the patch embeddings $\mathcal{E}_p$. Then, the patch embedding $\mathcal{E}_p$ is directly fed into the spatial-temporal Transformer to get the transformed embeddings $\mathcal{E}$. While the patch embeddings $\mathcal{E}_p$ are obtained using our original Patch Tokenizer (left part in Figure 1 (a)), we use a separate Target Patch Tokenizer (right part in Figure 1 (a)) for the target embeddings $\hat{\mathcal{E}}_p$ to provide self-supervision signals. The Target Patch Tokenizer is updated with an exponential moving average (EMA) of the original Patch Tokenizer weights. To encourage the model to learn functional context, the model is trained to discriminate replaced channels based on the spatial-temporal transformed embeddings $\mathcal{E} \in \{\boldsymbol{e}_{i,j} | i = 1, ..., C; j = 1, ..., N\}$, the target patch embeddings $\hat{\mathcal{E}}_p \in \{\hat{\boldsymbol{e}}_{i,j}^p | i = 1, ..., C; j = 1, ..., N\}$ and the replaced label $y \in \{-1, 1\}$, where $-1$ indicates unreplaced, 1 indicates replaced. The contrastive loss is defined as follows:

$$\mathcal{L}_{contra} = \sum_{i,j} \left[ 1 + y \cdot \langle \ell_2(\text{Linear}(\boldsymbol{e}_{i,j})), \ell_2(\hat{\boldsymbol{e}}_{i,j}^p) \rangle \right], \tag{4}$$

where $\ell_2$ represents $\ell_2$ normalization and $\langle \cdot, \cdot \rangle$ the inner product. Combined with $\ell_2$, $\langle \cdot, \cdot \rangle$ calculates the cosine similarity between $\boldsymbol{e}_{i,j}$ and $\hat{\boldsymbol{e}}_{i,j}^p$, which takes value within [-1,1]-range. $y$ is used to adjust whether to minimize or maximize such similarity, and the shift item 1 is added to ensure $\mathcal{L}_{contra} \geq 0$.

To further demonstrate the effectiveness of our proposed contrastive loss to capture functional context, we also design two alternatives to train the spatial-temporal model, i.e., $\mathcal{L}_{diff}$ and $\mathcal{L}_{mse}$. The difference loss $\mathcal{L}_{diff}$ also encourages maximizing the similarity between embeddings of unreplaced channels:

$$\mathcal{L}_{diff} = \sum_{i,j} \left[ \text{BCE}(\text{Linear}(||\boldsymbol{e}_{i,j} - \hat{\boldsymbol{e}}_{i,j}^p||_2^2)) \right], \tag{5}$$

where $|| \cdot ||_2^2$ represents the squared $\ell_2$ norm value and BCE represents the binary cross entropy loss.

Besides, following the commonly adopted JEPA pre-training framework (Assran et al., 2023; Dong et al., 2024), we randomly select patches to mask. Around 50% of patch embeddings $\mathcal{E}_p$ are patch-wise chosen and masked. The masked position is termed as $\mathcal{M}$. Then, a shared learnable embedding $\boldsymbol{e}_{[M]} \in \mathbb{R}^d$ is used to replace the original patch embeddings:

$$\mathcal{E}_m^f = \{\boldsymbol{e}_i^m | i = 1, ..., N\}, \quad \boldsymbol{e}_i^m = m_i \odot \boldsymbol{e}_{[M]} + (1 - m_i) \odot \boldsymbol{e}_i^p, \tag{6}$$

where $\delta(\cdot)$ is the indicator function and $m_i = \delta(i \in \mathcal{M})$. After that, the masked embeddings $\mathcal{E}_m$ will be fed into the spatial-temporal Transformer. The MSE loss $\mathcal{L}_{mse}$ is the average mean-squared error between spatial-temporal transformed embeddings $\mathcal{E}$ and target patch embeddings $\hat{\mathcal{E}}_p$:

$$\mathcal{L}_{mse} = \sum_{i,j \in \mathcal{M}} ||\boldsymbol{e}_{i,j} - \hat{\boldsymbol{e}}_{i,j}^p||_2^2. \tag{7}$$

## 3.3 DOWNSTREAM EVALUATION

We evaluate the validity of our training procedure and the effectiveness of our learned model using several downstream decoding tasks in the Brain Treebank dataset (Wang et al., 2024a). We also quantify whether BrainFCIR effectively models the functional context via channel clustering. First, we validate our pre-trained model's performance on four speech-perception related downstream tasks used by Wang et al. (2023); Chau et al. (2024): (1) Classification of low/high pitch; (2) Classification of low/high volume; (3) Identification of words that correspond to sentence onsets; (4) Classification of speech vs. non-speech audio. Classification performance is reported as an average across all hold-out test sessions (Appendix A), for 6 fine-tuning seeds each. As baselines, we compare our pre-trained model's fine-tuned performance against a fine-tuned, randomly-initialized version of itself, as well as two advanced spatial-temporal sEEG models: Population Transformer (PopT) (Chau et al., 2024) and Brant (Zhang et al., 2023). Second, we ablate the loss item used in the replaced functional

context discrimination task to explore the effectiveness of our proposed contrastive loss. Third, we quantify the effectiveness of functional context modeling via channel clustering, and visualize the identified functional groups to strengthen the neuroscientific interpretation. When pre-trained with the target subject, the selected channels via channel clustering further enhance downstream decoding; when evaluated on new subjects, our model demonstrates the great potential to offer an off-the-shelf functional group identification toolkit for unseen subjects. Lastly, towards the goal of building intracranial sEEG neurofoundation models, we evaluate our modeling framework's data scalability, overall model interpretability – the results of which are presented in Appendix E&I, respectively.

## 4 EXPERIMENTS

### 4.1 DATASET

For our experiments, we used the publicly available Brain Treebank dataset (Wang et al., 2024a), which consists of intracranial recordings (2048 Hz) from 10 epilepsy patients collected over a total of 26 sessions as they watched Hollywood films. Film transcripts that are aligned to neural activity are also provided. The intracranial sEEG recordings cover multiple brain regions across both hemispheres, including the temporal and frontal lobes, which are known to support auditory and language processing. Neural data is provided at a sampling rate of 2048 Hz. We followed a similar preprocessing procedure as outlined by Wang et al. (2023; 2024a); Chau et al. (2024).

We evaluate our model against previous advanced baselines on four binary classification tasks (e.g., sentence onset detection). Multi-channel sEEG signals are represented as $\mathcal{X} \in \mathbb{R}^{C \times T}$, and the paired label is $y \in \mathcal{Y}$, where $\mathcal{Y}$ represents the label-set. We use ROC-AUC as the evaluation metric.

### 4.2 IMPLEMENTATION DETAILS

**Preprocess.** The sEEG signals first undergo bandpass filtering between 0.5 Hz and 200 Hz to attenuate low-frequency drift and high-frequency noise. Following this, a 60 Hz notch filter is applied to suppress power-line interference. The signals are then resampled to 400 Hz and re-referenced (Li et al., 2018) according to the original setting to enhance the spatial resolution of the recordings. Finally, z-score normalization is applied independently to each channel to ensure consistent scaling across all channels, thereby facilitating stable model training.

**Model Configurations.** Throughout both pre-training and fine-tuning, raw input patches are initially tokenized into the patch embedding space with a dimensionality of $d = 256$. These embeddings are then processed by a sequence of transformer modules: first by a Temporal Transformer and subsequently by a Spatial Transformer. Each of these transformer modules is implemented as a 4-layer encoder block. The architecture maintains a consistent model dimension of $d = 256$ across layers, while the inner feed-forward network (FFN) dimension is expanded to $d_{ff} = 1024$. Each multi-head self-attention layer utilizes 8 parallel attention heads to capture diverse contextual relationships. A comprehensive breakdown of the model's hyperparameters and architectural specifics is provided in Appendix C.

**Pre-training.** The pre-training model is trained using all recordings across all subjects, excluding those reserved for validation and testing in downstream tasks (Appendix A). When using recordings from all subjects, the model is trained on 8 GPUs (NVIDIA Tesla V100 32GB using Python 3.11.7 and PyTorch 2.1.2 + CUDA 12.3) for ∼12 hours in total.

**Fine-tuning.** We split the task recordings into training, validation, and testing splits with a size roughly proportional to 80%, 10%, and 10%. All experiments are conducted on the same machine with the same set of random seeds. The train/validation/test splits are the same across different models. For each subject, models are trained for ∼20 minutes. The best models are trained on the training set, selected from the validation set according to accuracy, and finally evaluated on the test set. For model comparison, we report the average and standard error values (of all subjects) on six random seeds to obtain comparable results.

### 4.3 BRAINFCIR ENHANCES DECODING BY MODELING FUNCTIONAL CONTEXT

In Table 1, we report the average classification ROC-AUC over all test sessions and seeds. Our results demonstrate that our model outperforms all alternative models by effectively capturing the functional context. In comparison to randomly initialized versions of our model, our pre-training improves downstream performance. To further quantify our model's effectiveness in capturing functional context, we forward sEEG samples from the target subject into the frozen model to estimate the inter-channel functional connectivity (Appendix C). Based on the sparse inter-channel functional connectivity, we perform hard clustering ($k = 10$) to extract the functional groups. When evaluating the BrainFCIR model on the selected functional groups, our model further improves decoding performance while greatly reducing the inference time, as only $\sim$20% of channels are kept. Overall, the results in Table 1 demonstrate that by effectively modeling the functional context, our model can improve downstream task performance by learning functional-context-informed representations for multi-regional neural activity.

Table 1: Results on the Brain Treebank dataset, with mean ROC-AUC and s.e.m. reported. Asterisks indicate that the bolded model is significantly better than the second model ($p < 0.01$, paired T-test).

| Method | Pitch | Volume | Sentence Onset | Speech/Non-Speech |
|---|---|---|---|---|
| Brant | 0.61±0.03 | 0.74±0.03 | 0.80±0.04 | 0.80±0.03 |
| LaBraM | 0.69±0.03 | 0.83±0.02 | 0.87±0.02 | 0.85±0.02 |
| CBraMod | 0.71±0.03 | 0.86±0.03 | 0.88±0.01 | 0.87±0.02 |
| PopT | 0.74±0.03 | 0.87±0.03 | 0.90±0.01 | 0.93±0.02 |
| BrainFCIR | 0.77±0.02 | 0.89±0.02 | 0.94±0.01 | 0.96±0.01 |
| w/o pre-training | 0.59±0.03 | 0.71±0.05 | 0.80±0.03 | 0.79±0.05 |
| w/ channel-selection | **0.78±0.02**\* | **0.91±0.02**\* | **0.94±0.01** | **0.97±0.01**\* |

### 4.4 CONTRASTIVE LOSS DURING PRE-TRAINING ENHANCES DOWNSTREAM PERFORMANCE

We investigated how our proposed contrastive loss effectively models the functional context. To do so, we pre-trained our model using different loss items (including the original contrastive loss $\mathcal{L}_{contra}$, the difference loss $\mathcal{L}_{diff}$, and the JEPA-style (Assran et al., 2023) MSE loss $\mathcal{L}_{mse}$), which are detailed in Section 3.2. Then, we evaluated each pre-trained model's performance on the same speech perception tasks in Table 1.

First, we find that the choice of the pre-training objective has a substantial impact on downstream performance (Figure 2). Specifically, models trained with $\mathcal{L}_{contra}$ achieve the highest decoding accuracy, demonstrating the importance of explicitly modeling similarity relationships across neural states for capturing functional context. Second, $\mathcal{L}_{diff}$ underperforms relative to $\mathcal{L}_{contra}$, likely due to its sole reliance on differentiating samples without explicitly encouraging similarity among positive pairs, which appears critical for learning functionally meaningful representations. Third, $\mathcal{L}_{mse}$, which relies on a traditional reconstruction-based objective, yields the lowest performance. This suggests that an over-reliance on intra-channel temporal dynamics—without explicit inter-channel relational modeling—fails to capture the functional context necessary for robust speech perception decoding.

To statistically validate these observations, we performed paired T-tests for pairwise comparisons. The analysis revealed a significant main effect of the loss type. Post-hoc tests confirmed that $\mathcal{L}_{contra}$ significantly outperformed both $\mathcal{L}_{diff}$ ($p < 0.01$) and $\mathcal{L}_{mse}$ ($p < 0.001$), while $\mathcal{L}_{diff}$ also surpassed $\mathcal{L}_{mse}$ ($p < 0.001$). In summary, our results indicate that contrastive learning objectives – particularly those that balance similarity encouragement and dissimilarity constraints—are most effective for modeling functional context in neural signals, underscoring the importance of relational inductive biases in self-supervised pre-training for neural decoding.

### 4.5 BRAINFCIR CAN IDENTIFY FUNCTIONAL MODULES THROUGH CHANNEL CLUSTER

We further validated the neurobiological plausibility and generalization capability of BrainFCIR by visualizing its estimated functional connectivity (Figure 3) and examining how functional group selection generalizes across varying pre-training cohort sizes (Figure 4). We projected the inferred

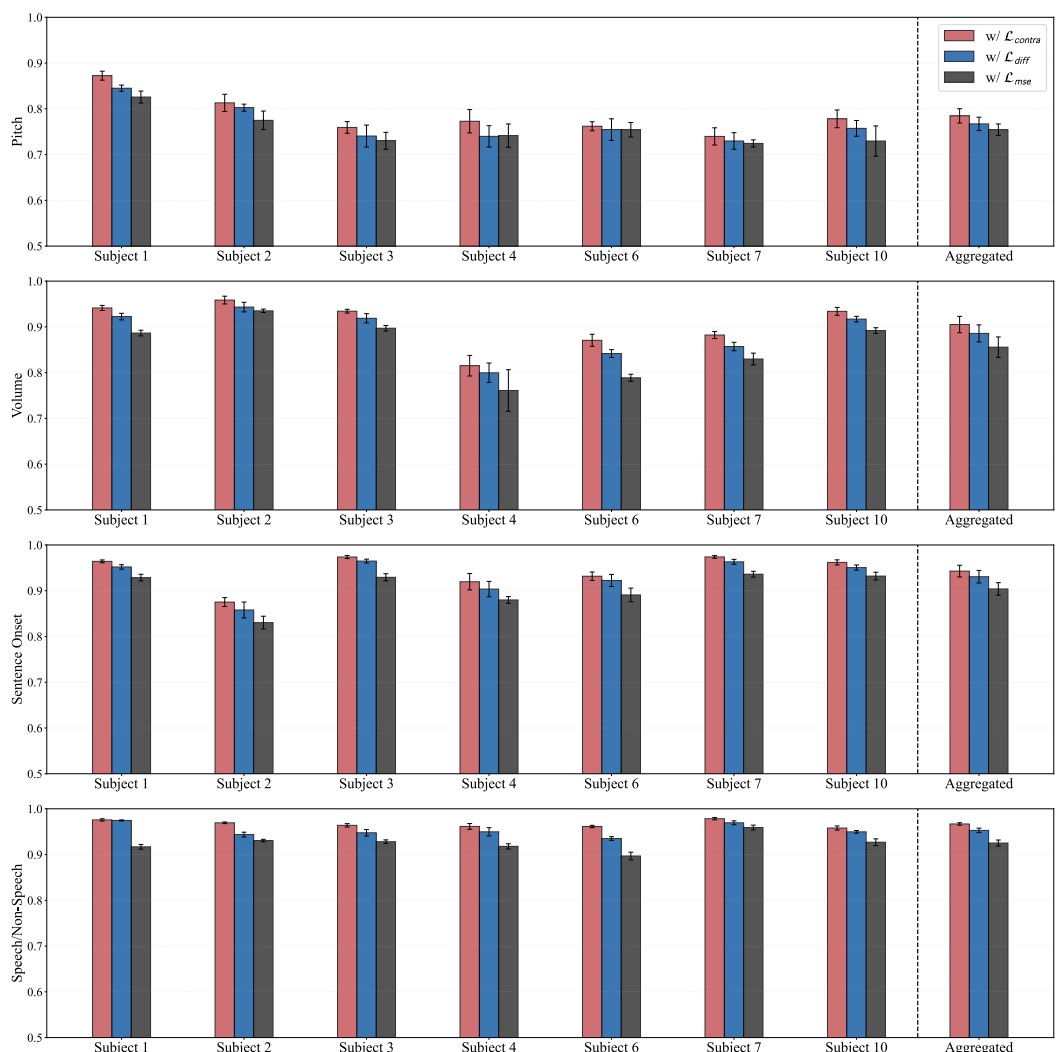

Figure 2: **Ablations on Functional Context Loss.** We pre-train BrainFCIR with different losses to encourage learning functional-context-informed representations. We report ROC-AUC for each task across 6 random seeds.

inter-channel functional connections onto cortical surface maps alongside the resulting functional clusters, enabling anatomical interpretation of the learned representations. Additionally, we systematically varied the number of subjects included in pre-training—with and without the target subject—to assess how cohort composition affects the quality of functionally-informed channel selection for downstream decoding.

First, we observed that the functional connectivity patterns estimated by BrainFCIR yield clusters that align well with known neuroanatomical regions (Figure 3), suggesting that the model captures biologically meaningful neural groupings. Second, when the pre-training dataset included only the target subject, channel selection based on the resulting model achieved the highest decoding accuracy, reflecting optimal adaptation to subject-specific functional organization. Third, introducing additional subjects initially reduced performance, likely due to interference from inter-subject variability; however, as more subjects were added, decoding accuracy gradually recovered and approached the single-subject baseline, indicating that the model learns to distill shared functional principles across individuals. Fourth, in the more challenging zero-shot setting where the target subject was excluded from pre-training, decoding performance improved steadily with larger pre-training cohorts, yet

consistently lagged behind the subject-included condition—highlighting a persistent gap attributable to individual-specific functional specializations.

These results demonstrate that while BrainFCIR captures generalizable functional topology, fully leveraging subject-specific functional specializations still requires target-subject data. Nevertheless, the model's ability to approach subject-specific performance with increasing cross-subject data underscores its potential for scalable neural decoding applications.

(a) Functional Clusters                               (b) Functional Connectivity

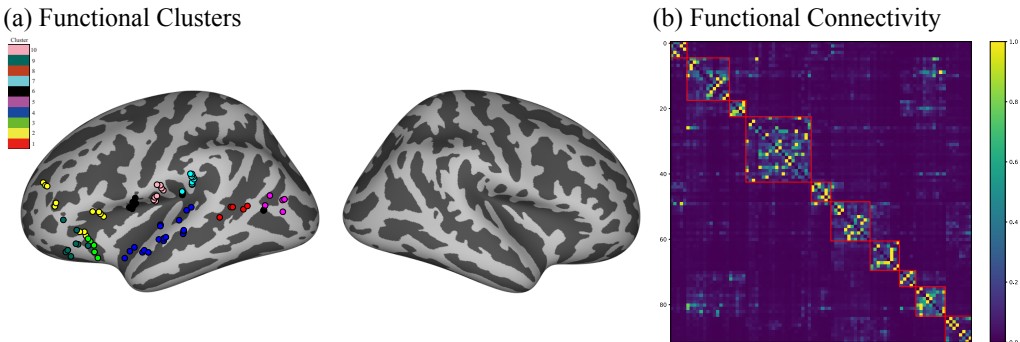

Figure 3: **Results for Functional Module Identification. (a).** The visualization of functional clusters identified by our method ($k = 10$). **(b).** The functional connectivity estimated by our method.

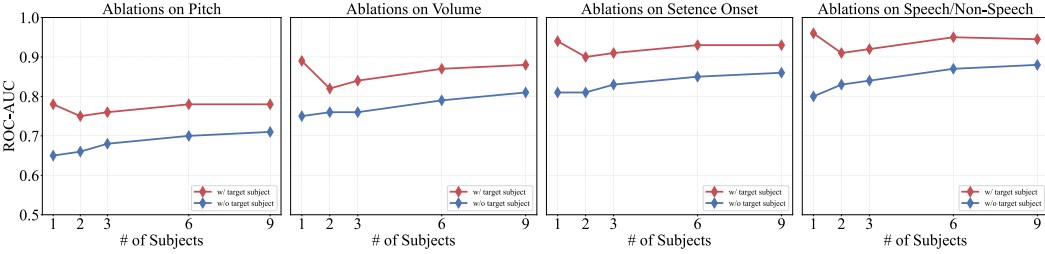

Figure 4: **Ablations on Channel Selection.** We pre-train BrainFCIR while varying the number of subjects (either w/ or w/o the target subject). The averaged ROC-AUC across subjects is reported.

## 5 DISCUSSION

Our work demonstrates that explicitly modeling functional context through spatial-context-guided representation learning significantly advances intracranial sEEG decoding. BrainFCIR not only achieves state-of-the-art performance but also produces functionally coherent channel clusters that align with neuroanatomy. The contrastive objective proves essential for capturing inter-channel relationships, outperforming reconstruction-based losses. Furthermore, while subject-specific pre-training yields optimal decoding, our model effectively generalizes functional topology across subjects, with performance scaling steadily with cohort size. These findings highlight that incorporating functional context is critical for building biologically plausible and high-performing neural decoders. Future work will explore dynamic functional networks and extend the framework to other cognitive domains.

## 6 CONCLUSION

This paper proposes BrainFCIR, a novel neurofoundation model for intracranial sEEG that explicitly incorporates functional context into representation learning via spatial-context-guided pre-training. Comprehensive experiments demonstrate that BrainFCIR not only achieves state-of-the-art performance in speech perception decoding on the Brain Treebank dataset, but also identifies functionally coherent channel clusters that align with known neuroanatomy. In addition, the model shows promising generalization in cross-subject functional group identification, with performance scaling robustly

as pre-training cohort size increases. Overall, our framework—informed by principles of brain network organization—provides a biologically interpretable and high-performing approach for neural decoding, moving toward more clinically applicable and transparent brain-computer interfaces.

## THE USAGE OF LLMS

Our writing process was assisted by DeepSeek-R1 (Guo et al., 2025), which was used to polish textual clarity. Brief paragraphs were provided to the model, and its output was critically evaluated before relevant revisions were adopted for the final version.

## REPRODUCIBILITY STATEMENT

Code to train models and reproduce the results will be publicly available. To facilitate review, an anonymous repository link will be provided during rebuttal stage, which is only visible to reviewers.

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

# A  TASK DETAILS

Brain Treebank (Wang et al., 2024a) dataset is a publicly available dataset of 10 epilepsy patients while they were watching movies from a set of 21 animated/action Hollywood movies. Each subject watched one or more movies while their brain activity was recorded (measured by sEEG). There is a total of 26 sessions across all subjects, each being ∼2 hours long on average.

## A.1  PRE-TRAINING DETAILS

We detail the pre-training configurations of BrainFCIR (Figure 1 (a)). From the 26 available sessions, 16 were used for training, 2 were held out as downstream validation, and the remaining 7 were held out for downstream testing, as specified in Table 2. We prepare the pre-training data by segmenting neural recordings for each session into non-overlapping intervals of 4 seconds, resulting in a total of 27,698 pre-training segments – corresponding to ∼30 hours.

Table 2: The session splits for pre-training in Brain Treebank.

| Subject | Session | Duration (hours) | Split |
|---------|---------|------------------|-------|
| Subject 1 | Session 1 | 1.91 | Train |
| | Session 2 | 2.90 | Test |
| | Session 3 | 2.07 | Train |
| Subject 2 | Session 1 | 2.60 | Train |
| | Session 2 | 2.42 | Train |
| | Session 3 | 2.66 | Train |
| | Session 4 | 3.00 | Train |
| | Session 5 | 3.73 | Train |
| | Session 6 | 1.85 | Valid |
| | Session 7 | 3.52 | Test |
| Subject 3 | Session 1 | 1.90 | Test |
| | Session 2 | 2.94 | Train |
| | Session 3 | 4.06 | Train |
| Subject 4 | Session 1 | 1.87 | Test |
| | Session 2 | 1.75 | Train |
| | Session 3 | 1.31 | Valid |
| Subject 5 | Session 1 | 1.54 | Train |
| Subject 6 | Session 1 | 0.81 | Train |
| | Session 2 | 1.32 | Train |
| | Session 3 | 1.60 | Test |
| Subject 7 | Session 1 | 1.67 | Test |
| | Session 2 | 1.77 | Train |
| Subject 8 | Session 1 | 1.41 | Train |
| Subject 9 | Session 1 | 1.00 | Train |
| Subject 10 | Session 1 | 1.57 | Test |
| | Session 2 | 2.33 | Train |

## A.2  FINE-TUNING DETAILS

We adopted the same task specification and analysis window in PopT (Chau et al., 2024), yielding 5-second neural activity per trial. We report the number of training, validation, and test trials for each downstream task in Table 3. The number of positive and negative labels is balanced.

**Pitch.**  The pitch of a given word is extracted using Librosa's `piptrack` function over a Mel-spectrogram (sampling rate 48,000 Hz, FFT window length of 2048, hop length of 512, and 128 mel

filters). For this task, for a given session, the positive examples consist of words in the top quartile of pitch, and the negative examples are the words in the bottom quartile.

**Volume.**   The volume of a given word is computed as the average intensity of root-mean-square (RMS) (`rms` function, frame and hop lengths 2048 and 512, respectively). For this task, for a given session, the positive examples are the words in the top quartile of volume, and the negative examples are the words in the bottom quartile.

**Sent. Onset (Sentence Onset).**   The negative examples are intervals of activity from 1s periods during which no speech occurs in the movie. The positive examples are intervals of brain activity that correspond with hearing the first word of a sentence.

**Word Onset (Speech vs. Non-speech).**   The negative examples are intervals of activity from 1s periods during which no speech occurs in the movie. The positive examples are intervals of brain activity that correspond with dialogue being spoken in the stimulus movie.

For each task, we follow the evaluation protocol in PopT (Chau et al., 2024), using the specified movie for downstream classification. Since these tasks are binary classification (CLS) tasks, we flatten embeddings and add a linear head after either pre-trained or randomly initialized models. Training employs binary cross-entropy (BCE) loss, with results quantified using ROC-AUC scores.

Table 3: The trial splits for fine-tuning in Brain Treebank.

| Subject | Pitch | | | Volume | | | Sent. Onset | | | Word Onset | | |
|---|---|---|---|---|---|---|---|---|---|---|---|---|
| | Train | Valid | Test | Train | Valid | Test | Train | Valid | Test | Train | Valid | Test |
| Subject 1 | 4076 | 510 | 510 | 4076 | 510 | 510 | 2358 | 318 | 318 | 10130 | 1267 | 1267 |
| Subject 2 | 2560 | 320 | 320 | 2560 | 320 | 320 | 1710 | 214 | 214 | 10236 | 1280 | 1280 |
| Subject 3 | 4038 | 505 | 505 | 4038 | 505 | 505 | 3282 | 411 | 411 | 4128 | 517 | 517 |
| Subject 4 | 996 | 125 | 125 | 996 | 125 | 125 | 866 | 109 | 109 | 3984 | 498 | 498 |
| Subject 6 | 2536 | 317 | 317 | 2536 | 317 | 317 | 1694 | 212 | 212 | 5092 | 637 | 637 |
| Subject 7 | 2932 | 367 | 367 | 2932 | 367 | 367 | 2068 | 259 | 259 | 4680 | 586 | 586 |
| Subject 10 | 3328 | 417 | 417 | 3328 | 417 | 417 | 2664 | 333 | 333 | 3786 | 474 | 474 |

## B   BASELINE DETAILS

In experiments, we compare our model to the existing advanced neurofoundation models (Zhang et al., 2023; Chau et al., 2024) on intracranial sEEG signals. The details of these baseline models are given here:

- **Brant** (Zhang et al., 2023): A self-supervised model for sEEG recordings that can capture both long-term temporal dependency and spatial correlation from neural signals. Brant is primarily designed for medical use, serving as an sEEG foundation model. Although Brant mainly evaluates its performance on the low-level modeling tasks (Wu et al., 2022) (e.g., neural signal forecasting, imputation, etc.), Brant achieves SOTA performance on some high-level modeling tasks (e.g., seizure detection). As a foundation model in the sEEG pre-training field, this model is suitable to serve as a baseline for comparison.

- **PopT** (Chau et al., 2024): A self-supervised model for sEEG that learns population-level codes for arbitrary ensembles of neural recordings at scale. PopT stacks on top of pre-trained temporal embeddings (Wang et al., 2023) and enhances downstream decoding by enabling the learned aggregation of multiple spatially sparse channels. PopT serves as an sEEG foundation model, achieving SOTA performance on Brain Treebank (Wang et al., 2024a). As a foundation model in the sEEG pre-training field, this model is suitable to serve as a baseline for comparison.

The detailed implementations of these baseline models are given here:

- For the Brant method (Zhang et al., 2023), the hyperparameters are optimized based on the Brant-Tiny model for better performance. We changed the length of the patch segment from 6 seconds to 1 second. Additionally, we replace the linear embedding layer with a convolutional embedding layer, which is used in LaBraM (Jiang et al., 2024b). The numbers of convolution filters are $\{96, 96, 96\}$; the sizes of convolution kernels are $\{9, 9, 3\}$; the numbers of convolution strides are $\{5, 5, 1\}$.

- For the PopT method (Chau et al., 2024), the hyperparameters are the same as the original implementation of the PopT model. The data samples are resampled to the specified sampling rate (i.e., 2048 Hz).

When evaluating the decoding performance of these baseline models, we follow the same experiment setup as our model; see Appendix C for more details.

For the self-supervised methods, the pre-training setup follows the original setup of each model:

- For the Brant model, we also use all sEEG recordings from all subjects within the Brain Treebank dataset to pre-train it. While the total pre-training dataset is smaller than the one used in the original paper, the number of subjects (i.e., the number of sEEG location configurations) is greater than in the original paper. The data samples are 4 seconds.

- For the PopT model, we include neural recordings from all available subjects within the Brain Treebank dataset for pre-training. The data samples are 4 seconds.

## C  MODEL DETAILS

The BrainFCIR model (Table 4) is a novel neurofoundation model for intracranial sEEG recordings, as shown in Figure 1 (a). The architecture of BrainFCIR contains two parts: (1) Patch Tokenizer, (2) Temporal & Spatial Transformer, and (3) Channel Cluster Module. During the pre-training stage, one additional "Token Predictor" (i.e., linear projection) is added after the "Spatial Transformer" for functional context discrimination.

**Functional Context Discrimination.**   Since sEEG channels capture local and depth information from different brain regions, their recordings inherently capture unique neural information with minimal overlap. This makes the functional context discrimination task better suited for learning inter-channel relationships compared to mask-based reconstruction approaches (Zhang et al., 2023; Jiang et al., 2024b). To ensure balanced label distribution, we designate only 10% of unreplaced channels as positive samples during pre-training.

**Channel Cluster Module.**   After pre-training with the spatial context task, we calculate the channel connectivity $\mathcal{P} \in \mathbb{R}^{C \times C}$ following Algorithm 1. Then, spectral cluster (Ng et al., 2001) is applied to group channels into functional clusters, using scikit-learn's (Pedregosa et al., 2011) `cluster.SpectralClustering` with default function arguments.

---

**Algorithm 1** The calculation of channel connectivity $\mathcal{P} \in \mathbb{R}^{C \times C}$.

---

**Require:** $\{\mathcal{X}_i \in \mathbb{R}^{C \times T} | i = 1, ..., N_{\text{samples}}\}$     $\triangleright N_{\text{samples}}$ is the number of samples.
  $\mathcal{P} \leftarrow \mathbf{0}_{C \times C}$                     $\triangleright \mathcal{P} \in \mathbb{R}^{C \times C}$ is initialized as 0s.
  **while** $i \leq N_{\text{samples}}$ **do**
    $\hat{\mathcal{P}} \leftarrow \text{model}(\mathcal{X}_i)$          $\triangleright \hat{\mathcal{P}} \in \mathbb{R}^{N_{\text{layer}} \times N_{\text{head}} \times C \times C}$ is spatial attention scores.
    $\hat{\mathcal{P}} \leftarrow \text{mean}(\hat{\mathcal{P}}, \text{axes} = [0, 1])$   $\triangleright \hat{\mathcal{P}} \in \mathbb{R}^{C \times C}$ is averaged across [layer,head]-dimensions.
    $\mathcal{P} \leftarrow \mathcal{P} + \hat{\mathcal{P}}/N_{\text{samples}}$
  **end while**

---

## D  MODEL EFFICIENCY

Table 5 shows the FLOPs (with `thop` package) and per-trial inference time across all methods. Our model achieves superior efficiency with the smallest parameter count (3.32M) and lower computational footprint (19.13 GFLOPs), enabling faster inference (23.73 ms) compared to existing approaches. With functional channel selection, computational cost drops significantly to 2.42 GFLOPs while maintaining—or even improving—decoding performance, further reducing inference time to 16.37 ms. These results demonstrate that our method offers a highly efficient and practical solution for decoding speech perception from intracranial sEEG recordings. All experiments were conducted on a single NVIDIA V100 GPU.

## E  DATA SCALING

To evaluate data efficiency, we assessed downstream classification performance (Table 6) of BrainF-CIR when pretrained on progressively larger fractions of the available data (5% to 75%). Performance demonstrated a clear scaling trend with increased pretraining data. Data subsets were constructed through incremental session-wise addition until the target percentage was met. This process was repeated across 6 random seeds to ensure robustness. For smaller data fractions, we adjusted the number of pretraining epochs to maintain a consistent total number of parameter updates.

## F  ABLATIONS ON CHANNEL SELECTION

The spatial-attention weights are not equivalent to traditional signal-level functional connectivity (FC) measures (e.g., corherence). The averaged spatial-attention weights can be viewed as an alternative to

Table 4: The hyperparameters for BrainFCIR training.

| Module | Name | Value |
|---|---|---|
| Patch Tokenizer | # of Input Channels | {1,64,64} |
| | # of Output Channels | {64,64,64} |
| | Kernel Size | {9,9,3} |
| | Stride | {5,5,2} |
| | Padding | {4,4,1} |
| | Flatten Window | 2 |
| Temporal Transformer | # of Transformer Layers | 4 |
| | Hidden Size | 128 |
| | MLP Size | 512 |
| | MLP Dropout Ratio | {0.2,0.} |
| | # of Attention Heads | 8 |
| | Attention Head Size | 64 |
| | Attention Dropout Ratio | 0.2 |
| Spatial Transformer | # of Transformer Layers | 4 |
| | Hidden Size | 128 |
| | MLP Size | 512 |
| | MLP Dropout Ratio | {0.2,0.} |
| | # of Attention Heads | 8 |
| | Attention Head Size | 64 |
| | Attention Dropout Ratio | 0.2 |
| Token Predictor | Linear Projection | $128 \rightarrow 128$ |
| Optimizer | Batch Size | 64 |
| | Maximum Learning Rate | 3e-4 |
| | Minimum Learning Rate | 5e-6 |
| | Learning Rate Scheduler | Cosine |
| | Optimizer Type | AdamW |
| | Adam $\beta$ | (0.9, 0.99) |
| | Weight Decay | 0.05 |
| | Total Epochs | 100 |
| | Warm-up Epochs | 10 |
| | EMA momentum schedule | linear |
| | EMA start momentum | 0.996 |
| | EMA final momentum | 1 |

Table 5: Model Efficiency Analysis on Brain Treebank dataset.

| Methods | Model Size | GFLOPs | Time (ms) |
|---|---|---|---|
| Brant | 500M | 116.2700±6.6485 | 54.11±5.21 |
| PopT | 20M | 27.9417±1.7639 | 26.39±3.16 |
| BrainFCIR | **3.32M** | 19.1308±1.0945 | 23.73±2.92 |
| w/ channel-selection | - | **2.4153±0.1719** | **16.37±2.19** |

analyze connectivity. To further illustrate the advantages of our spatial-attention-based estimation of functional connectivity, we additionally perform clustering on the corherence-based FC, ranking each cluster using a downstream task. Since channel selection based on BrainFCIR estimation only uses channels contained in the first cluster, we also report the decoding performance on the first cluster to evaluate whether the corherence-based FC clustering can accurately identify functional boundaries.

Since corherence primarily estimates channel connectivity based on low-order correlations between channels, it is more susceptible to the influence of channel anatomical proximity. Compared to our

Table 6: BrainFCIR's downstream performance scales as a function of pre-training data size.

| Data Percentage | Pitch | Volume | Sentence Onset | Speech/Non-Speech |
|---|---|---|---|---|
| 100% | 0.77±0.02 | 0.89±0.02 | 0.94±0.01 | 0.96±0.01 |
| 75% | 0.76±0.02 | 0.87±0.02 | 0.93±0.01 | 0.95±0.01 |
| 50% | 0.75±0.02 | 0.88±0.02 | 0.93±0.02 | 0.95±0.01 |
| 25% | 0.71±0.03 | 0.83±0.04 | 0.88±0.04 | 0.90±0.03 |
| 10% | 0.68±0.04 | 0.80±0.03 | 0.85±0.05 | 0.87±0.03 |
| 5% | 0.64±0.04 | 0.75±0.04 | 0.83±0.04 | 0.83±0.05 |

Table 7: BrainFCIR's downstream performance varies across different connectivity estimation for channel selection.

| Connectivity Type | Pitch | Volume | Sentence Onset | Speech/Non-Speech |
|---|---|---|---|---|
| - | 0.77±0.02 | 0.89±0.02 | 0.94±0.01 | 0.96±0.01 |
| BrainFCIR | **0.78±0.02** | **0.91±0.02** | **0.94±0.01** | **0.97±0.01** |
| Corherence | 0.75±0.02 | 0.88±0.02 | 0.90±0.01 | 0.92±0.01 |

method (Table 7), it struggles to effectively estimate the precise boundaries of functional modules, resulting in lower performance.

Besides, we further execute ablations on the number of clusters (Table 8). Fewer clusters (e.g., 5) reduce the spatial resolution of functional groups and may include irrelevant channels. More clusters maintain performance but require more group-level evaluation and combination.

Table 8: BrainFCIR's downstream performance maintains when varying number of clusters for channel selection.

| # of clusters | 5 | 8 | 10 | 15 | 20 |
|---|---|---|---|---|---|
| Pitch | 0.77±0.02 | 0.78±0.02 | 0.78±0.02 | 0.78±0.02 | 0.78±0.02 |
| Volume | 0.90±0.02 | 0.90±0.02 | 0.91±0.02 | 0.91±0.02 | 0.91±0.02 |
| Sentence Onset | 0.94±0.01 | 0.94±0.01 | 0.94±0.01 | 0.94±0.01 | 0.94±0.01 |
| Speech/Non-Speech | 0.96±0.01 | 0.97±0.01 | 0.97±0.01 | 0.97±0.01 | 0.97±0.01 |

## G Cross-subject Transfer

To evaluate the generalizability of our pretrained weights, we performed a leave-one-out (LOO) cross-validation. A model was pretrained on all but one subject, then fine-tuned and evaluated on the held-out subject. Results indicate that excluding a subject from pretraining does not significantly impact downstream performance (Table 9), demonstrating the robustness and potential utility of our approach for new, unseen data.

## H Additional Results

Since Brain Treebank dataset was collected while subjects watched movies, we extracted the audio portion of the movies to evaluate speech synthesis tasks (Chen et al., 2024). The evaluation results of different baselines are shown in Table 10. Our model still outperforms all baselines, demonstrating that the explicit modeling of functional connections helps in decoding cognitive tasks such as speech perception. We reported the Pearson Correlation Coefficient between the predicted mel-spectrogram and the ground truth.

Table 9: BrainFCIR's downstream performance in leave-one-out (LOO) setting.

| Setting | Pitch | Volume | Sentence Onset | Speech/Non-Speech |
|---------|-------|--------|----------------|-------------------|
| - | 0.77±0.02 | 0.89±0.02 | 0.94±0.01 | 0.96±0.01 |
| LOO | 0.74±0.03 | 0.87±0.02 | 0.93±0.02 | 0.94±0.01 |

Table 10: BrainFCIR's downstream performance on speech synthesis task.

| Model | Brant | LaBraM | CBraMod | PopT | BrainFCIR |
|-------|-------|--------|---------|------|-----------|
| PCC | 0.6571±0.0104 | 0.6491±0.0121 | 0.6784±0.0109 | 0.7017±0.0105 | **0.7228±0.0115** |

## I MODEL INTERPRETABILITY

To visualize electrode coverage across brain regions (Figure 5), we mapped intracranial electrode locations to anatomical regions using the Desikan-Killiany atlas (Desikan et al., 2006). For each subject, electrode coordinates were registered to the fsaverage surface template and assigned to corresponding cortical parcellations. ROI activation intensities were computed by normalizing electrode counts per region across subjects and experimental conditions. The resulting intensity maps were projected onto inflated cortical surfaces using Nilearn's surface plotting functions. Brain visualizations display both hemispheres in lateral view with a color-coded intensity scale (red colormap) representing normalized electrode density, providing clear spatial representation of recording coverage across cortical areas for each experimental task.

We performed channel clustering analysis across four distinct auditory-linguistic tasks: (1) Pitch; (2) Volume; (3) Sentence Onset; (4) Speech/Non-Speech. Based on final classification performance, we selected specific cluster groups and analyzed the spatial distribution of channels within these clusters. The clustering results revealed two distinct patterns. Pitch and volume classification tasks demonstrated consistent clustering patterns, while sentence onset detection and speech/non-speech classification tasks showed similar groupings to each other but differed from the pitch/volume conditions. For pitch and volume classification tasks, selected channels were predominantly distributed in bilateral auditory regions, specifically the superior temporal gyrus and transverse temporal gyrus. Secondary distributions were observed in Wernicke's area and the middle frontal gyrus, suggesting engagement of both primary auditory processing and higher-order linguistic regions.

In contrast, sentence onset detection and speech/non-speech classification tasks showed channels primarily concentrated in the same bilateral auditory areas (superior temporal gyrus and transverse temporal gyrus), with comparable representation in Wernicke's area. However, a striking difference emerged in the middle frontal gyrus, where virtually no channels were selected for these tasks, distinguishing them from the pitch/volume conditions. This differential pattern suggests distinct neural mechanisms underlying continuous acoustic feature processing versus discrete linguistic event detection, which is consistent with the results of previous neuroscience research (DeWitt & Rauschecker, 2013; Friederici, 2011; Van der Burght et al., 2019).

(a) Pitch & Volume

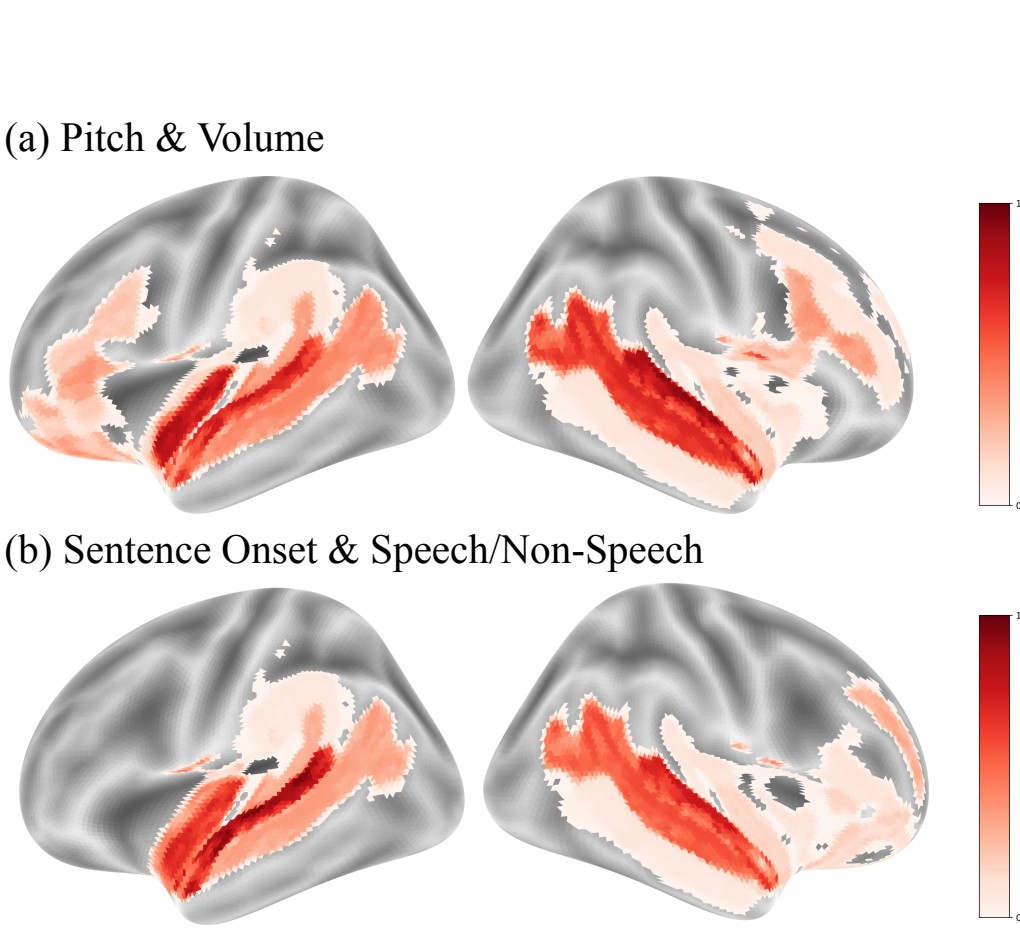

(b) Sentence Onset & Speech/Non-Speech

Figure 5: Visualization of the alignment between selected groups and language-related regions.

## J    SUBJECT-WISE CHANNEL CLUSTER

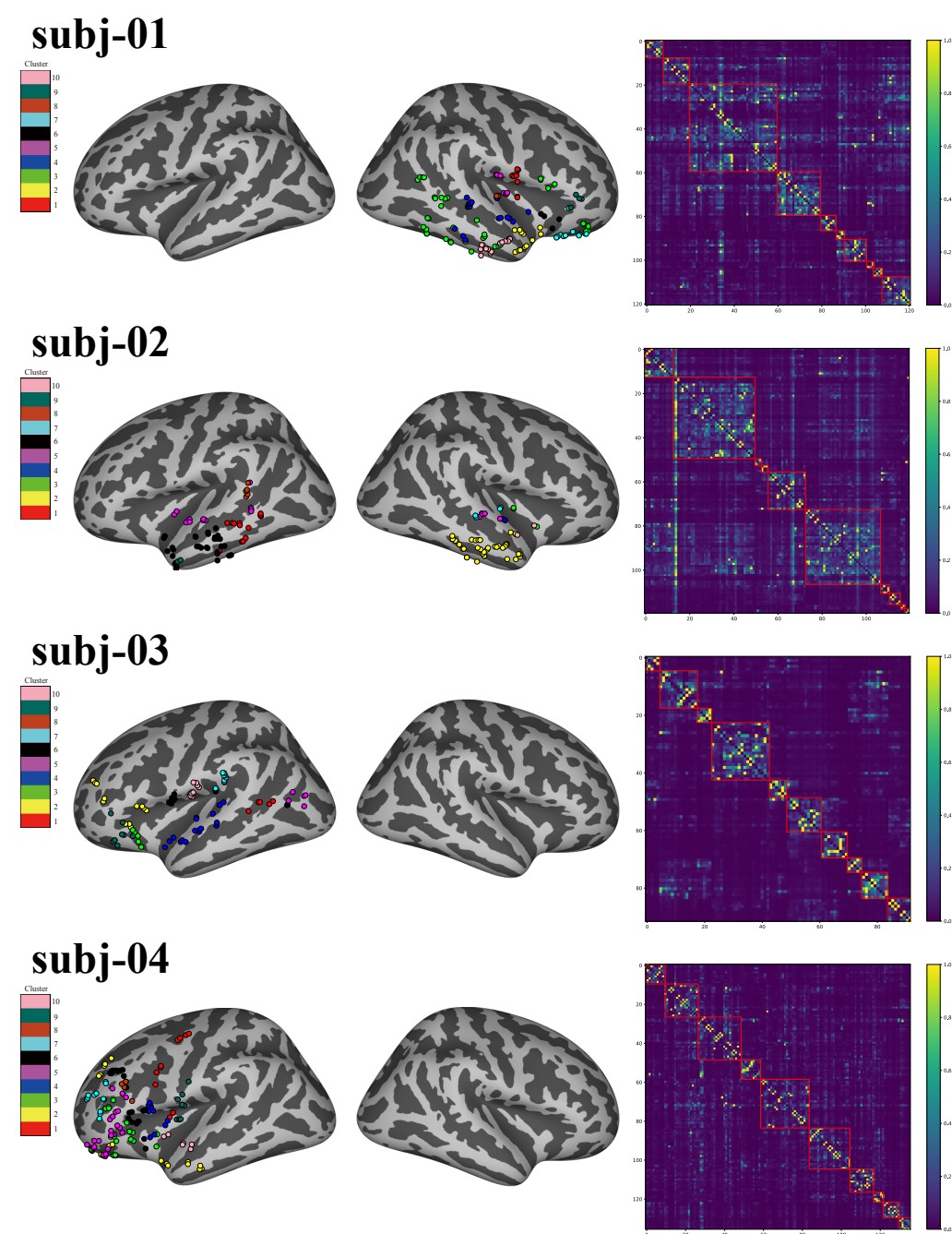

Figure 6: Channel clusters from subjects (01-04).

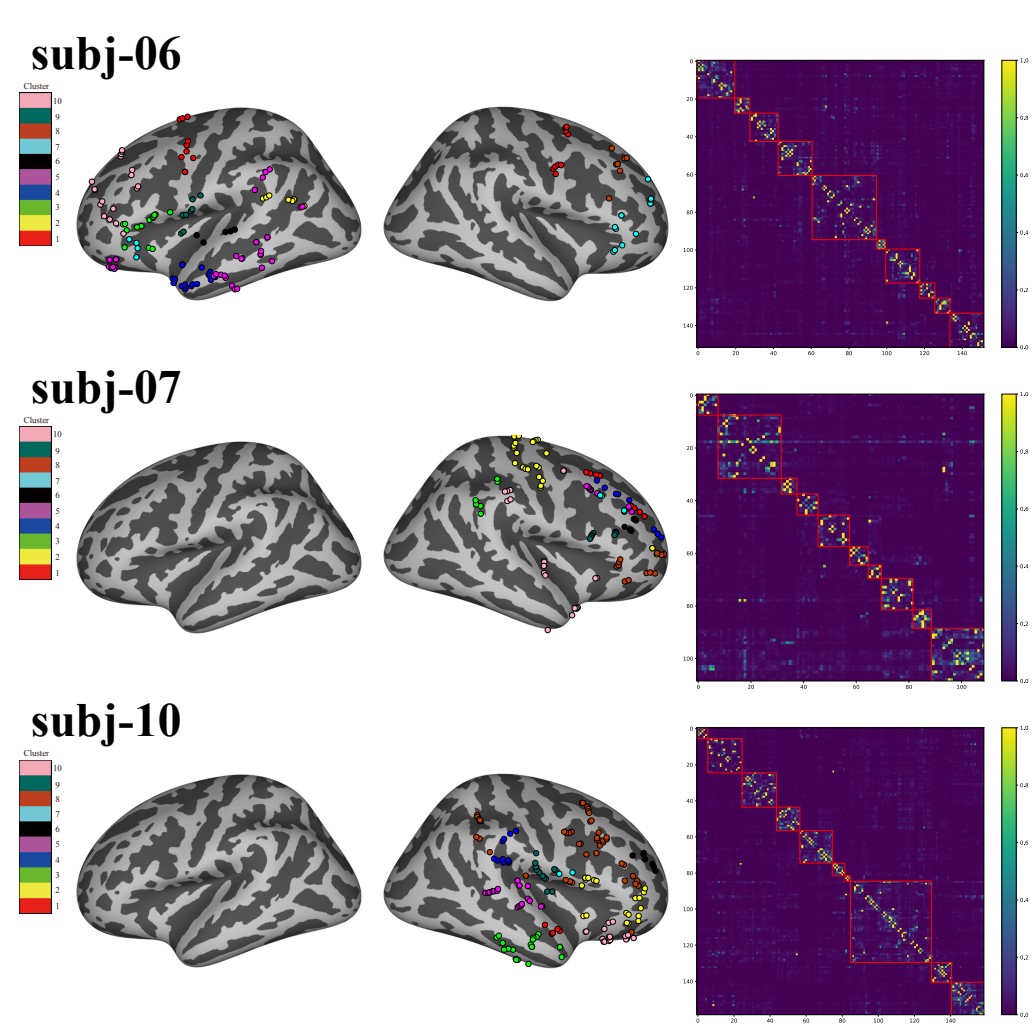

Figure 7: Channel clusters from subjects (05,06,10).

