# OpenReview forum: "BrainFCIR: Functional Context Informed Representation Learning for Intracranial Neural Signals"
_ICLR.cc/2026/Conference — Submitted to ICLR 2026_

### Official Review · Reviewer_JhKT · 2025-10-15

**Soundness:** 4
**Presentation:** 3
**Contribution:** 4
**Rating:** 8
**Confidence:** 3

**Summary:**

In contrast with foundation models that learn temporal or spatial representations of sEEG signals, the proposed model BrainFCIR represents the spatio-temporal relationships. This is done by learning at the same time the tokenization of the signal and a representation based on a spatio-temporal transformer approach. In addition to providing an unified representation, the model also outperforms current decoding methods.

**Strengths:**

the model is original and addresses an important topic, both in terms of representation learning and in terms of performance.

**Weaknesses:**

The topic of building representations integrating contextual constraints is central in this work. Here the context is related to responses of the neighborhood, but it is not clear (and not discussed) if other kind of constraints are absent in the current SOTA and should be also considered here.

**Questions:**

-the need to have a mapping that also integrates contextual information is also present in many other domains than sEEG. Do you think your work could be applied to other signals ?
- the database corresponds to recording of 10 different people. Do you think that it is enough or that better performance could be observed with more people ?

---

> ### Author Response · Authors · 2025-11-22
> **Response to Reviewer JhKT by Authors (1/2)**
>
> Thank you for the thorough review and constructive comments. We are deeply grateful for acknowledging our method “is original and addresses an important topic, both in terms of representation learning and in terms of performance”. Please see below for the point-by-point responses to your comments.
>
> # Reply to Weaknesses
>
> > **W1**: The topic of building representations integrating contextual constraints is central in this work. Here the context is related to responses of the neighborhood, but it is not clear (and not discussed) if other kind of constraints are absent in the current SOTA and should be also considered here.
>
> **R**: Thank you for the valuable feedback. In clinical settings, the implantation sites of sEEG electrodes vary significantly among subjects, essentially representing a partial recording of the brain. This work aims to **integrate data from different subjects through a pre-training task focused on spatial context discrimination, to better understand the functional connections between various recording sites within the brain**, thereby learning a unified and generalizable representation to enhance downstream decoding tasks. Furthermore, we analyze the structure of the model's attention map and evaluate its ability to accurately identify functional group boundaries using the decoding performance of different clusters after clustering, thus validating its interpretability. Therefore, in this process, **“responses of the neighborhood” emerge spontaneously through pre-training, consistent with the modular organization of the brain**. Previous pre-training methods (such as Brant [1] and BrainWave [2]) primarily use mask reconstruction, and these models do not introduce additional constraints. However, because these pre-training tasks over-rely on the intra-channel temporal patterns, they may slightly weaken the capture of functional connections between different sites, resulting in slightly lower decoding performance.
>
> **References**:
>
> [1] Zhang D, Yuan Z, Yang Y, et al. Brant: Foundation model for intracranial neural signal[J]. Advances in Neural Information Processing Systems, 2023, 36: 26304-26321.
>
> [2] Yuan Z, Shen F, Li M, et al. Brainwave: A brain signal foundation model for clinical applications[J]. arXiv preprint arXiv:2402.10251, 2024.
>
> # Reply to Questions
>
> > **Q1**: The need to have a mapping that also integrates contextual information is also present in many other domains than sEEG. Do you think your work could be applied to other signals ?
>
> **R**: Thank you for the insightful feedback. Our method's ability to identify the brain's modular organization [1] (Figure 3) capitalizes on the key advantage of intracranial sEEG: its capacity to provide highly localized, non-redundant neural signals. This allows us to probe functional organization at a significantly higher spatial resolution. We anticipate direct generalization to ECoG data, given their shared characteristic of recording localized neural populations.
>
> Furthermore, we believe the core principles of our model are applicable to non-invasive modalities like EEG/MEG. While these signals exhibit greater noise and channel coupling, the fundamental nature of underlying neural computation is conserved. Our approach is designed to learn dynamic functional connectivity from data; we therefore anticipate that it can distill meaningful functional relationships even from non-invasive recordings, thereby providing a unified framework to enhance decoding across a spectrum of neural data types.
>
> **References**:
>
> [1] Silva A B, Littlejohn K T, Liu J R, et al. The speech neuroprosthesis[J]. Nature Reviews Neuroscience, 2024, 25(7): 473-492.

---

> > ### Author Response · Authors · 2025-11-22
> > **Response to Reviewer JhKT by Authors (2/2)**
> >
> > > **Q2**: The database corresponds to recording of 10 different people. Do you think that it is enough or that better performance could be observed with more people ?
> >
> > **R**: Thank you for the valuable feedback. We agree that a larger sample size may improve the model's performance and generalization ability. While our current dataset of 10 subjects is consistent with previous sEEG studies [1,2], it does have limitations. Figure 4 and Appendix E demonstrate that BrainFCIR's performance is positively correlated with the amount of pre-training data, indicating that increasing the number of subjects further improves recognition and decoding accuracy across subject functional groups. Nevertheless, even with only 10 subjects, our model achieves state-of-the-art performance and demonstrates good cross-subject generalization ability in both channel selection and neural decoding (Table 9 in the updated version). We believe that as more sEEG data becomes available, this will strongly support the scalability and practical application potential of BrainFCIR.
> >
> > In addition, since Brain Treebank dataset was collected while subjects watched movies, we extracted the audio portion of the movies to evaluate speech synthesis tasks [3]. The evaluation results of different baselines are as follows. Our model still outperforms all baselines, demonstrating that the explicit modeling of functional connections helps in decoding cognitive tasks such as speech perception. We reported the Pearson Correlation Coefficient between the predicted mel-spectrogram and the ground truth.
> >
> > | Model | Brant | LaBraM | CBraMod | PopT | BrainFCIR |
> > | --- | --- | --- | --- | --- | --- |
> > | - | 0.6571$\pm$0.0104 | 0.6491$\pm$0.0121 | 0.6784$\pm$0.0109 | 0.7017$\pm$0.0105 | **0.7228$\pm$0.0115** |
> >
> > **References**:
> >
> > [1] Chau G, Wang C, Talukder S, et al. Population transformer: Learning population-level representations of neural activity[J]. ArXiv, 2025: arXiv: 2406.03044 v4.
> >
> > [2] Wang C, Yaari A, Singh A, et al. Brain treebank: Large-scale intracranial recordings from naturalistic language stimuli[J]. Advances in Neural Information Processing Systems, 2024, 37: 96505-96540.
> >
> > [3] Chen X, Wang R, Khalilian-Gourtani A, et al. A neural speech decoding framework leveraging deep learning and speech synthesis[J]. Nature Machine Intelligence, 2024, 6(4): 467-480.

---

### Official Review · Reviewer_HkRQ · 2025-10-31

**Soundness:** 2
**Presentation:** 2
**Contribution:** 2
**Rating:** 2
**Confidence:** 2

**Summary:**

This paper introduces BrainFCIR, a neurofoundation model for intracranial sEEG recordings that aims to model functional context (e.g. inter-channel relationships reflecting coordinated brain activity). The model is pretrained with constrastive learning and combines temporal and spatial transformers from which they estimate inter-channel connectivity. Then, they cluster channels into functional groups that are finally used for decoding and interpretability. The results they provide on the Brain Treebank human sEEG dataset show improvements over Brant and PopT on four speech related decoding tasks. The proposed architecture and pretraining objective are technically sound and the evaluation protocol follows established practice. However the experiments use a limited number of seeds, comparison are made with only two baselines, and quantitative gains are moderate. While the method appears valid, stronger statistical validation and more comparative results would increase confidence in the claims.

**Strengths:**

BrainFCIR successfully unifies spatial and temporal modeling within a contrastive pre-training framework to identify functional connectivity patterns in the brain using transformer attention matrices. The model achieves modest but consistent improvements over the compared baselines on sEEG decoding tasks. The paper is well structured, conceptually coherent, and relies on publicly available data, which supports reproducibility and transparency.

**Weaknesses:**

The results, while promising, remain preliminary. The experimental validation is limited to four tasks and only two baselines; additional benchmarks would help position the contribution. The qualitative interpretability analysis lacks depth and justification (e.g., why only one hemisphere, how clusters map to known regions). Figures are difficult to read (notably text and legend are very small), captions are minimal, and the code is not yet released. Finally, while computational gains from channel selection are mentioned, no quantitative comparison supports the claim.

**Questions:**

1. What is the rationale for showing only one hemisphere in the cortical visualizations ?
2. Could you provide quantitative results on inference speed and efficiency after channel selection ?
3. Why are there only two baselines for comparison? Please either clarify the reasons (e.g., lack of other available models) or include more baselines in the evaluation.
4. Why is the code not available with the submission? This raises concerns about the reproducibility of the work.

---

> ### Author Response · Authors · 2025-11-22
> **Response to Reviewer HkRQ by Authors (1/2)**
>
> Thank you for the thorough review and constructive comments. We are deeply grateful for acknowledging our method “achieves modest but consistent improvements over the compared baselines on sEEG decoding tasks”. Please see below for the point-by-point responses to your comments.
>
> # Reply to Weaknesses
>
> > **W1**: The experimental validation is limited to four tasks and only two baselines; additional benchmarks would help position the contribution.
>
> **R**: Thank you for the helpful suggestion. To make a more comprehensive comparison, we introduce two additional EEG foundation model baselines: LaBraM [1], and CBraMod [2]. Their results on the Brain Treebank dataset are reported below:
>
> | Task | Pitch | Volume |  Sentence Onset | Speech/Non-Speech |
> | --- | --- | --- | --- | --- |
> | LaBraM | 0.69$\pm$0.03 | 0.83$\pm$0.02 | 0.87$\pm$0.02 | 0.85$\pm$0.02 |
> | CBraMod | 0.71$\pm$0.03 | 0.86$\pm$0.03 | 0.88$\pm$0.01 | 0.87$\pm$0.02 |
>
> To enable LaBraM to be pre-trained across subjects on intracranial sEEG recordings, we replaced its learnable positional encoding, which was adapted for specific EEG cap configurations, with the positional encoding based on MNI coordinates that we used. Since these models are primarily pre-trained for mask reconstruction, as discussed in Line 59, they may be insufficient in modeling the functional relationships between channels, resulting in slightly worse performance compared to the PopT baseline.
>
> In addition, since Brain Treebank dataset was collected while subjects watched movies, we extracted the audio portion of the movies to evaluate speech synthesis tasks [3]. The evaluation results of different baselines are as follows. Our model still outperforms all baselines, demonstrating that the explicit modeling of functional connections helps in decoding cognitive tasks such as speech perception. We reported the Pearson Correlation Coefficient between the predicted mel-spectrogram and the ground truth.
>
> | Model | Brant | LaBraM | CBraMod | PopT | BrainFCIR |
> | --- | --- | --- | --- | --- | --- |
> | - | 0.6571$\pm$0.0104 | 0.6491$\pm$0.0121 | 0.6784$\pm$0.0109 | 0.7017$\pm$0.0105 | **0.7228$\pm$0.0115** |
>
> Due to the current lack of publicly available datasets in this field, the amount of data evaluated is consistent with previous works [4,5]. If more datasets become available, we also hope that our model can be extended to more cognitive decoding tasks (such as image perception).
>
> **References**:
>
> [1] Jiang W B, Zhao L M, Lu B L. Large brain model for learning generic representations with tremendous EEG data in BCI[J]. arXiv preprint arXiv:2405.18765, 2024.
>
> [2] Wang J, Zhao S, Luo Z, et al. Cbramod: A criss-cross brain foundation model for eeg decoding[J]. arXiv preprint arXiv:2412.07236, 2024.
>
> [3] Chen X, Wang R, Khalilian-Gourtani A, et al. A neural speech decoding framework leveraging deep learning and speech synthesis[J]. Nature Machine Intelligence, 2024, 6(4): 467-480.
>
> [4] Chau G, Wang C, Talukder S, et al. Population transformer: Learning population-level representations of neural activity[J]. ArXiv, 2025: arXiv: 2406.03044 v4.
>
> [5] Wang C, Yaari A, Singh A, et al. Brain treebank: Large-scale intracranial recordings from naturalistic language stimuli[J]. Advances in Neural Information Processing Systems, 2024, 37: 96505-96540.

---

> > ### Author Response · Authors · 2025-11-22
> > **Response to Reviewer HkRQ by Authors (2/2)**
> >
> > > **W2**: The qualitative interpretability analysis lacks depth and justification (e.g., why only one hemisphere, how clusters map to known regions).
> >
> > **R**: Thank you for the helpful suggestion. Since the implantation location of sEEG is mainly determined by the location of the subject's epilepsy, the subject we are showing has electrodes implanted only in the left brain. Based on the “Desikan-Killiany” anatomical template, we calculate the weighted anatomy region counts $N_{a}$ for different channel cluster methods to provide quantitative comparisons. Specifically, $N_{a}=\sum\frac{C_{i}}{C}N_{a}^{i}$, where $C_{i}$ is the number of channels within that cluster, $C$ is the total number of channels, $N_{a}^{i}$ is the anatomy region counts of that cluster. A value of $N_{a}=1$ indicates perfect alignment with anatomical regions, while $N_{a}$ approaching the total number of regions suggests limited functional grouping capability. Thus, lower $N_{a}$ values reflect stronger anatomical alignment in channel clustering. We reported the metric over the clustering results of coherence-based functional connectivity and our method.
> >
> > | Method | Coherence | BrainFCIR |
> > | --- | --- | --- |
> > | Brain Treebank | 7.52$\pm$1.26 | **4.38$\pm$0.40** |
> >
> > > **W3**: Figures are difficult to read (notably text and legend are very small), captions are minimal, and the code is not yet released. Finally, while computational gains from channel selection are mentioned, no quantitative comparison supports the claim.
> >
> > **R**: Thank you for the helpful suggestion. We modified the font size of all axis labels, legends, and annotations in the Figures, with more detailed descriptions to facilitate a thorough examination. We have every intention to release code after manuscript acceptance as stated in our abstract.
> >
> > As for the computational gains from channel selection, please refer to Table 5 in Appendix D, which provides a comprehensive quantitative analysis. The table shows that our full BrainFCIR model requires only 19.13 GFLOPs, which is substantially lower than Brant (116.27 GFLOPs) and PopT (27.94 GFLOPs). When applying channel selection, the computational cost plummets to 2.42 GFLOPs. This is an order-of-magnitude reduction from both our own full model and the baselines, while simultaneously improving ROC-AUC (as shown in Table 1 of the main text).
> >
> > # Reply to Questions
> >
> > > **Q1**: What is the rationale for showing only one hemisphere in the cortical visualizations ?
> >
> > **R**: Thank you for the valuable feedback. Please refer to **Reply to W2** for more details.
> >
> > > **Q2 & Q4**: Could you provide quantitative results on inference speed and efficiency after channel selection ? Why is the code not available with the submission? This raises concerns about the reproducibility of the work.
> >
> > **R**: Thank you for the valuable feedback. Please refer to **Reply to W3** for more details.
> >
> > > **Q3**: Why are there only two baselines for comparison? Please either clarify the reasons (e.g., lack of other available models) or include more baselines in the evaluation.
> >
> > **R**: Thank you for the helpful suggestion. Please refer to **Reply to W1** for more details.

---

### Official Review · Reviewer_medq · 2025-11-01

**Soundness:** 3
**Presentation:** 3
**Contribution:** 3
**Rating:** 4
**Confidence:** 4

**Summary:**

This paper introduces BrainFCIR, a spatiotemporal transformer for sEEG that is pre-trained using a “functional-context replacement” task and later used for channel clustering and selection. The idea is to encourage the model to capture cross-channel dependencies, and then exploit the resulting attention structure to identify informative channels. On several speech-related classification tasks, BrainFCIR reports modest gains over prior baselines and reduced inference cost after channel selection.

**Strengths:**

- The concept of treating functional context as a self-supervised signal is creative and potentially useful for building population-level representations from intracranial data.
- Leveraging the learned attention maps for channel selection is a nice practical touch — if validated properly, it could be valuable for faster or more portable brain–computer interfaces.

**Weaknesses:**

- Ambiguity in the pretraining task. It’s not clear exactly how the replaced segments are sampled (from the same session? same channel? matched by power spectrum?). The teacher/student inputs and the label definitions in Eq. (4–5) are hard to follow, and the objective could be solved by simple statistical shortcuts rather than genuine contextual reasoning.

- Questionable interpretation of attention as connectivity. Averaging spatial-attention weights is not equivalent to functional connectivity, especially since anatomical coordinate embeddings are also injected. The resulting maps could mostly reflect geometric proximity. There’s no comparison to standard signal-level FC measures (e.g., coherence or partial correlation) or distance-controlled analyses.

- Unclear clustering visualizations. In Fig. 3, the spatial distribution is hard to interpret (why only one hemisphere? why uneven density?). The matrix view in panel (b) lacks meaningful axes or clear correspondence to panel (a). The number of clusters k=10 is fixed without sensitivity analysis.

- Experimental results:  Weak cross-subject generalization. Under the hold-one-subject-out setting, BrainFCIR drops more sharply than PopT (≈ 0.10 vs 0.05 gap). Multi-subject training barely improves performance, suggesting poor subject scalability.

- Possible data leakage in channel selection. If the “functional connectivity” used for clustering is computed on the entire dataset (even unlabeled), this would constitute double-dipping and inflate test scores.

- Baseline alignment. Preprocessing (sampling rate, referencing) and hyperparameter changes make comparisons with PopT and Brant hard to interpret. Also, model selection is by accuracy while reporting uses AUC — these should match.

- Parameter inconsistency. The reported 3.32 M parameters don’t match the configuration in Table 4; just the transformer MLPs would already exceed that count.

- Limited task diversity. Only binary classification is shown. Including a regression or retrieval task (e.g., continuous speech features or EEG image reconstruction [1]) would better demonstrate representational value.

**Questions:**

- how are replacement segments chosen and normalized?

- What input does the teacher see?

- Is channel selection computed strictly on training data per fold?

- How does attention-derived connectivity relate to Euclidean distance or to coherence-based FC?

- Can you reconcile the parameter count with a detailed component breakdown?

---

> ### Author Response · Authors · 2025-11-22
> **Response to Reviewer medq by Authors (1/5)**
>
> Thank you for the thorough review and constructive comments. We are deeply grateful for the acknowledgment that our method “is creative and potentially useful for building population-level representations from intracranial data”. Please see below for the point-by-point responses to your comments.
>
> # Reply to Weaknesses
> > **W1**: Ambiguity in the pretraining task. It’s not clear exactly how the replaced segments are sampled (from the same session? same channel? matched by power spectrum?). The teacher/student inputs and the label definitions in Eq. (4–5) are hard to follow, and the objective could be solved by simple statistical shortcuts rather than genuine contextual reasoning.
>
> **R**: Thank you for the valuable feedback. To clarify, the replaced segments are sampled from different time points (could be different sessions) but the same channel. Since all sEEG signals are z-score normalized per channel during preprocessing, the replaced segments have similar low-level statistics (zero mean and unit variance) to the original data. This design is intentional to eliminate simple intra-channel statistical cues (e.g., mean or variance shifts) as shortcuts. Instead, the model must rely on the multivariate functional context -- i.e., whether the activity of a channel is coherent with the simultaneous activity of all other channels in the current sample. By replacing a channel with data from a different temporal context (same channel, different time points), we force the model to learn inter-channel dependencies and functional relationships, which are essential for robust representation learning.
>
> Our task design specifically mitigates this issue: **because replaced segments come from the same channel (same statistics) and are z-score normalized, the model cannot rely on intra-channel properties like mean or variance**. Instead, it must exploit inter-channel contextual relationships -- e.g., synchrony or functional connectivity -- to detect inconsistencies. The contrastive loss (Eq. 4) reinforces this by comparing embeddings across channels.
>
> > **W2**: Questionable interpretation of attention as connectivity. Averaging spatial-attention weights is not equivalent to functional connectivity, especially since anatomical coordinate embeddings are also injected. The resulting maps could mostly reflect geometric proximity. There’s no comparison to standard signal-level FC measures (e.g., coherence or partial correlation) or distance-controlled analyses.
>
> **R**: Thank you for the insightful feedback. We agree that spatial-attention weights are not direct proxies for traditional functional connectivity (FC) metrics. Instead, they provide a powerful, dynamic alternative for analyzing functional interactions. While anatomical embeddings offer a static spatial prior, **our attention mechanism dynamically modulates channel interactions based on real-time neural activity, thereby capturing context-dependent relationships that transcend fixed anatomical proximity**.
>
> To empirically validate this advantage, we conducted a comparative analysis: we clustered channels using coherence-based FC and ranked the clusters via downstream task performance. We then evaluated the BrainFCIR’s decoding performance using only the top-ranked cluster, mirroring our method's evaluation.
>
> | Connectivity Type | Pitch | Volume | Sentence Onset | Speech/Non-Speech |
> | --- | --- | --- | --- | --- |
> | Corherence | 0.75$\pm$0.02 | 0.88$\pm$0.02 | 0.90$\pm$0.01 | 0.92$\pm$0.01 |
>
> The results confirm **a key limitation of coherence-based FC: by relying on low-order signal correlations, it remains heavily influenced by anatomical proximity**. Consequently, it fails to delineate precise functional boundaries as effectively as our method, leading to the lower performance we report. Our approach, in contrast, infers functional organization from the model's spatial attention, yielding a more accurate and functionally relevant grouping.
>
> > **W3**: Unclear clustering visualizations. In Fig. 3, the spatial distribution is hard to interpret (why only one hemisphere? why uneven density?). The matrix view in panel (b) lacks meaningful axes or clear correspondence to panel (a). The number of clusters k=10 is fixed without sensitivity analysis.
>
> **R**: Thank you for the valuable feedback. The placement of sEEG electrodes is solely determined by clinical need, which varies significantly across epilepsy patients. This results in highly subject-specific implantation configurations with uneven channel density -- as exemplified by the subject shown, who has electrodes only in the left hemisphere. Consequently, decoding performance can vary substantially across subjects, which is precisely why channel selection is a critical and standard preprocessing step in the field [1-4] to remove irrelevant signals and boost performance, especially for complex tasks such as word classification in [3].

---

> > ### Author Response · Authors · 2025-11-22
> > **Response to Reviewer medq by Authors (2/5)**
> >
> > Furthermore, we conducted an ablation study on the number of clusters. BrainFCIR’s downstream performance maintains when varying the number of clusters for channel selection. We found that too few clusters (e.g., 5) reduces spatial resolution and can dilute group functionality by including irrelevant channels. Conversely, while more clusters maintain high performance, they increase the computational overhead for group-level evaluation. Our chosen parameter thus represents an optimal balance.
> >
> > | # of clusters | 5 | 8 | 10 | 15 | 20 |
> > | --- | --- | --- | --- | --- | --- |
> > | Pitch | 0.77$\pm$0.02 | 0.78$\pm$0.02 | 0.78$\pm$0.02 | 0.78$\pm$0.02 | 0.78$\pm$0.02 |
> > | Volume | 0.90$\pm$0.02 | 0.90$\pm$0.02 | 0.91$\pm$0.02 | 0.91$\pm$0.02 | 0.91$\pm$0.02 |
> > | Sentence Onset | 0.94$\pm$0.01 | 0.94$\pm$0.01 | 0.94$\pm$0.01 | 0.94$\pm$0.01 | 0.94$\pm$0.01 |
> > | Speech/Non-Speech | 0.96$\pm$0.01 | 0.97$\pm$0.01 | 0.97$\pm$0.01 | 0.97$\pm$0.01 | 0.97$\pm$0.01 |
> >
> > **References**:
> >
> > [1] Mentzelopoulos G, Chatzipantazis E, Ramayya A G, et al. Neural decoding from stereotactic EEG: accounting for electrode variability across subjects[J]. Advances in Neural Information Processing Systems, 2024, 37: 108600-108624.
> >
> > [2] Chau G, Wang C, Talukder S, et al. Population transformer: Learning population-level representations of neural activity[J]. ArXiv, 2025: arXiv: 2406.03044 v4.
> >
> > [3] Zheng H, Wang H, Jiang W, et al. Du-IN: Discrete units-guided mask modeling for decoding speech from Intracranial Neural signals[J]. Advances in Neural Information Processing Systems, 2024, 37: 79996-80033.
> >
> > [4] Wu D, Li S, Feng C, et al. Towards Homogeneous Lexical Tone Decoding from Heterogeneous Intracranial Recordings[J]. arXiv preprint arXiv:2410.12866, 2024.
> >
> > > **W4**: Experimental results: Weak cross-subject generalization. Under the hold-one-subject-out setting, BrainFCIR drops more sharply than PopT (0.10 vs 0.05 gap). Multi-subject training barely improves performance, suggesting poor subject scalability.
> >
> > **R**: Thank you for the valuable feedback. There may be some potential misunderstandings regarding Figure 4. Figure 4 mainly shows the impact of using BrainFCIR for cross-subject channel selection with the number of subjects used in pre-training. The result shows that, without using the target subject data, **pre-training on more subjects allows the model to learn more general functional connections across subjects, thereby more accurately estimating functional module boundaries on unseen subjects**, thus improving decoding performance through channel selection. To eliminate the concerns related to **the neural decoding's leave-one-subject-out setting**, we execute additional experiments to evaluate our method in the leave-one-out setting (LOO).
> > | Task | Pitch | Volume | Sentence Onset | Speech/Non-Speech |
> > | --- | --- | --- | --- | --- |
> > | BrainFCIR (LOO) | 0.74$\pm$0.03 | 0.87$\pm$0.02 | 0.93$\pm$0.02 | 0.94$\pm$0.01 |
> >
> > > **W5**: Possible data leakage in channel selection. If the “functional connectivity” used for clustering is computed on the entire dataset (even unlabeled), this would constitute double-dipping and inflate test scores.
> >
> > **R**: Thank you for the helpful suggestion. After clustering channels into functional groups based on the inter-channel attention graph learned via pure self-supervision, the channel groups are fixed. To avoid potential information leakage from the task dataset, we only use the unlabeled part for calculating channel connectivity. Under this evaluation, the selected channels are almost identical to those selected using all the data. This is because **clustering the estimated channel functional connections primarily extracts functional modules, not functional networks**. Due to the modular organization of the brain, even without task-state data, functional modules, as the basic organizational modules, can still be accurately identified for downstream channel selection.
> >
> > > **W6**: Baseline alignment. Preprocessing (sampling rate, referencing) and hyperparameter changes make comparisons with PopT and Brant hard to interpret. Also, model selection is by accuracy while reporting uses AUC — these should match.
> >
> > **R**: Thank you for the helpful suggestion. The re-referencing procedure strictly adheres to the original implementation in PopT (Line 299). As the Brant model was not designed for Laplacian-referenced data, we retrained it on the native, unreferenced sEEG signals. We find that modeling this raw data is advantageous, as the inherent channel correlations provide a proxy for anatomical proximity. This effectively compensates for Brant's lack of explicit spatial coordinate inputs, ultimately leading to the observed performance improvement.
> >
> > | Task | Pitch | Volume | Sentence Onset | Speech/Non-Speech |
> > | --- | --- | --- | --- | --- |
> > | Brant (un-referenced) | 0.64$\pm$0.03 | 0.79$\pm$0.04 | 0.84$\pm$0.03 | 0.82$\pm$0.03 |

---

> > > ### Author Response · Authors · 2025-11-22
> > > **Response to Reviewer medq by Authors (3/5)**
> > >
> > > To address potential misalignment from sampling rate differences, we resampled the sEEG signals to 2000 Hz (aligning with PopT's 2048 Hz) and 240 Hz (aligning with Brant's 250 Hz). The hyperparameters for the Patch Tokenizer were tuned accordingly for each rate. The results demonstrate that our method's performance is robust to these variations, showing no significant deviation from the results achieved with the original 400 Hz data.
> > >
> > > | Task | Pitch | Volume | Sentence Onset | Speech/Non-Speech |
> > > | --- | --- | --- | --- | --- |
> > > | BrainFCIR (2000Hz) | 0.77$\pm$0.02 | 0.89$\pm$0.02 | 0.94$\pm$0.02 | 0.96$\pm$0.01 |
> > > | BrainFCIR (240Hz) | 0.77$\pm$0.02 | 0.89$\pm$0.02 | 0.94$\pm$0.01 | 0.96$\pm$0.01 |
> > >
> > > When resampling sEEG signals to 2000Hz, the hyperparameters are modified as follows:
> > >
> > > | Module | Name | Value |
> > > | --- | --- | --- |
> > > | | Kernel Size | {49,9,3} |
> > > | Patch Tokenizer | Stride | {25,5,2} |
> > > | | Padding | {24,4,1} |
> > >
> > > When resampling sEEG signals to 240Hz, the hyperparameters are modified as follows:
> > >
> > > | Module | Name | Value |
> > > | --- | --- | --- |
> > > | | Kernel Size | {9,5,3} |
> > > | Patch Tokenizer | Stride | {5,3,2} |
> > > | | Padding | {4,2,1} |
> > >
> > > > **W7**: Parameter inconsistency. The reported 3.32 M parameters don’t match the configuration in Table 4; just the transformer MLPs would already exceed that count.
> > >
> > > **R**: Thank you for the valuable feedback. We are deeply sorry for any confusion caused by model hyperparameters in Table 4. Due to our oversight, we incorrectly reported the number of convolutional kernels in the Tokenizer section, leading to errors in subsequent parameters. We have corrected the model hyperparameter settings in Table 4. The detailed component breakdown is provided in “Reply to Q5”. **We sincerely apologize again for this and have carefully checked other details of the article to ensure that errors caused by human negligence are avoided.** Thank you again for your careful review.
> > >
> > > > **W8**: Limited task diversity. Only binary classification is shown. Including a regression or retrieval task (e.g., continuous speech features or EEG image reconstruction [1]) would better demonstrate representational value.
> > >
> > > **R**: Thank you for the helpful suggestion. To make a more comprehensive comparison, we introduce two additional EEG foundation model baselines: LaBraM [1], and CBraMod [2]. Their results on the Brain Treebank dataset are reported below:
> > > | Task | Pitch | Volume |  Sentence Onset | Speech/Non-Speech |
> > > | --- | --- | --- | --- | --- |
> > > | LaBraM | 0.69$\pm$0.03 | 0.83$\pm$0.02 | 0.87$\pm$0.02 | 0.85$\pm$0.02 |
> > > | CBraMod | 0.71$\pm$0.03 | 0.86$\pm$0.03 | 0.88$\pm$0.01 | 0.87$\pm$0.02 |
> > >
> > > To enable LaBraM to be pre-trained across subjects on intracranial sEEG recordings, we replaced its learnable positional encoding, which was adapted for specific EEG cap configurations, with the positional encoding based on MNI coordinates that we used. Since these models are primarily pre-trained for mask reconstruction, as discussed in Line 59, they may be insufficient in modeling the functional relationships between channels, resulting in slightly worse performance compared to the PopT baseline.
> > >
> > > In addition, since Brain Treebank dataset was collected while subjects watched movies, we extracted the audio portion of the movies to evaluate speech synthesis tasks [3]. The evaluation results of different baselines are as follows. Our model still outperforms all baselines, demonstrating that the explicit modeling of functional connections helps in decoding cognitive tasks such as speech perception. We reported the Pearson Correlation Coefficient between the predicted mel-spectrogram and the ground truth.
> > >
> > > | Model | Brant | LaBraM | CBraMod | PopT | BrainFCIR |
> > > | --- | --- | --- | --- | --- | --- |
> > > | - | 0.6571$\pm$0.0104 | 0.6491$\pm$0.0121 | 0.6784$\pm$0.0109 | 0.7017$\pm$0.0105 | **0.7228$\pm$0.0115** |
> > >
> > > Due to the current lack of publicly available datasets in this field, the amount of data evaluated is consistent with previous works [4,5]. If more datasets become available, we also hope that our model can be extended to more cognitive decoding tasks (such as image perception).

---

> > > > ### Author Response · Authors · 2025-11-22
> > > > **Response to Reviewer medq by Authors (4/5)**
> > > >
> > > > **References**:
> > > >
> > > > [1] Jiang W B, Zhao L M, Lu B L. Large brain model for learning generic representations with tremendous EEG data in BCI[J]. arXiv preprint arXiv:2405.18765, 2024.
> > > >
> > > > [2] Wang J, Zhao S, Luo Z, et al. Cbramod: A criss-cross brain foundation model for eeg decoding[J]. arXiv preprint arXiv:2412.07236, 2024.
> > > >
> > > > [3] Chen X, Wang R, Khalilian-Gourtani A, et al. A neural speech decoding framework leveraging deep learning and speech synthesis[J]. Nature Machine Intelligence, 2024, 6(4): 467-480.
> > > >
> > > > [4] Chau G, Wang C, Talukder S, et al. Population transformer: Learning population-level representations of neural activity[J]. ArXiv, 2025: arXiv: 2406.03044 v4.
> > > >
> > > > [5] Wang C, Yaari A, Singh A, et al. Brain treebank: Large-scale intracranial recordings from naturalistic language stimuli[J]. Advances in Neural Information Processing Systems, 2024, 37: 96505-96540.
> > > >
> > > > # Reply to Questions
> > > >
> > > > > **Q1**: How are replacement segments chosen and normalized?
> > > >
> > > > **R**: Thank you for the valuable feedback. Please see “Reply to W1” for more details.
> > > >
> > > > > **Q2**: What input does the teacher see?
> > > >
> > > > **R**: Thank you for the valuable feedback. If we understand it correctly, does "teacher" here refer to the target Patch Tokenizer that updates parameters via EMA? Crucially, both the student and teacher models receive the same sEEG input following the replacement operation.
> > > >
> > > > The core mechanism is as follows: when a channel's neural activity is replaced, its resulting embedding -- after being processed and contextualized by the spatiotemporal Transformer -- deviates from the stable representation generated by the target teacher tokenizer. This deliberate discrepancy directly creates the negative sample contrast, as formalized in Equation (4).
> > > >
> > > > > **Q3**: Is channel selection computed strictly on training data per fold?
> > > >
> > > > **R**: Thank you for the valuable feedback. After clustering channels into functional groups based on the inter-channel attention graph learned via pure self-supervision, the channel groups are fixed. We directly train and evaluate neural decoding model by randomly selecting 600 samples from train splits to rank channel clusters. Here, we demonstrate the ranked cluster-wise performance (w/ 600 samples) on the Brain Treebank (sentence onset detection) dataset. We can see that even with limited label data, we are still able to determine which clusters the task information is primarily encoded in due to the modular brain computation.
> > > >
> > > > | Cluster Index | 1 | 2 | 3 | 4 | 5 | 6 | 7 | 8 | 9 | 10 |
> > > > | --- | --- | --- | --- | --- | --- | --- | --- | --- | --- | --- |
> > > > | Brain Treebank | 0.86$\pm$0.03 | 0.67$\pm$0.02 | 0.62$\pm$0.02 | 0.61$\pm$0.02 | 0.58$\pm$0.01 | 0.57$\pm$0.02 | 0.56$\pm$0.01 | 0.56$\pm$0.02 | 0.55$\pm$0.01 | 0.54$\pm$0.01 |
> > > >
> > > > In practice, **although not strictly limiting channel selection to training data may lead to information leakage, based on the great performance difference among channel groups, this evaluation strategy results in the same selected channels and downstream performance**.
> > > >
> > > > > **Q4**: How does attention-derived connectivity relate to Euclidean distance or to coherence-based FC?
> > > >
> > > > **R**: Thank you for the valuable feedback. Please see “Reply to W2” for more details.
> > > >
> > > > > **Q5**: Can you reconcile the parameter count with a detailed component breakdown?
> > > >
> > > > **R**: Thank you for the valuable feedback. We are deeply sorry for any confusion caused by model hyperparameters in Table 4. Due to our oversight, we incorrectly reported the number of convolutional kernels in the Tokenizer section, leading to errors in subsequent parameters. We have corrected the model hyperparameter settings in Table 4. The detailed component breakdown by `print(model)` is as follows:

---

> > > > > ### Author Response · Authors · 2025-11-22
> > > > > **Response to Reviewer medq by Authors (5/5)**
> > > > >
> > > > > ```
> > > > > =======================================================================================================================================
> > > > > Layer (type:depth-idx)                                       Param #                   Param %                   Trainable
> > > > > =======================================================================================================================================
> > > > > brainfcir_ctx                                                    --                             --                   Partial
> > > > > ├─PatchTokenizer: 1-1                                        --                             --                   True
> > > > > │    └─Sequential: 2-1                                       --                             --                   --
> > > > > │    │    └─LambdaLayer: 3-1                                 --                             --                   --
> > > > > │    └─ModuleList: 2-2                                       --                             --                   True
> > > > > │    │    └─Sequential: 3-2                                  640                         0.02%                   True
> > > > > │    └─Sequential: 2-3                                       --                             --                   True
> > > > > │    │    └─Sequential: 3-3                                  --                             --                   --
> > > > > │    │    └─Sequential: 3-4                                  37,056                      1.11%                   True
> > > > > │    │    └─Sequential: 3-5                                  37,056                      1.11%                   True
> > > > > │    │    └─Sequential: 3-6                                  12,480                      0.38%                   True
> > > > > │    │    └─Sequential: 3-7                                  --                             --                   --
> > > > > │    └─Sequential: 2-4                                       --                             --                   --
> > > > > │    │    └─LambdaLayer: 3-8                                 --                             --                   --
> > > > > ├─TimeEmbedding: 1-2                                         (2,048)                     0.06%                   False
> > > > > ├─TransformerStack: 1-3                                      --                             --                   Partial
> > > > > │    └─ModuleList: 2-5                                       --                             --                   Partial
> > > > > │    │    └─TransformerBlock: 3-9                            404,641                    12.16%                   Partial
> > > > > │    │    └─TransformerBlock: 3-10                           404,641                    12.16%                   Partial
> > > > > │    │    └─TransformerBlock: 3-11                           404,641                    12.16%                   Partial
> > > > > │    │    └─TransformerBlock: 3-12                           404,641                    12.16%                   Partial
> > > > > ├─ChannelEmbedding: 1-4                                      --                             --                   --
> > > > > │    └─Sequential: 2-6                                       --                             --                   --
> > > > > │    │    └─LambdaLayer: 3-13                                --                             --                   --
> > > > > │    │    └─LambdaLayer: 3-14                                --                             --                   --
> > > > > ├─TransformerStack: 1-5                                      --                             --                   Partial
> > > > > │    └─ModuleList: 2-7                                       --                             --                   Partial
> > > > > │    │    └─TransformerBlock: 3-15                           404,641                    12.16%                   Partial
> > > > > │    │    └─TransformerBlock: 3-16                           404,641                    12.16%                   Partial
> > > > > │    │    └─TransformerBlock: 3-17                           404,641                    12.16%                   Partial
> > > > > │    │    └─TransformerBlock: 3-18                           404,641                    12.16%                   Partial
> > > > > ├─Sequential: 1-6                                            --                             --                   True
> > > > > │    └─TokenCLSHead: 2-8                                     --                             --                   True
> > > > > │    │    └─Sequential: 3-19                                 129                         0.00%                   True
> > > > > │    └─LambdaLayer: 2-9                                      --                             --                   --
> > > > > =======================================================================================================================================
> > > > > Total params: 3,326,537
> > > > > Trainable params: 3,324,481
> > > > > Non-trainable params: 2,056
> > > > > =======================================================================================================================================
> > > > > ```

---

### Meta-Review · Area_Chair_MMth · 2026-01-17

**Summary:**

`medq` provides the most complete review, wherein they provide a list of both positives (echoed by the other reviewers) and negatives. In particular, experimental concerns are voiced, both in limited scope and in its procedure. `HkRQ` has similar concerns about the experimental breadth, though somewhat ameliorated by author response.

Given two negative responses and the brevity of the third response, I am inclined to reject this manuscript.

**Reviewer Concerns:**

In general I feel that experimental concerns are partially unaddressed. Further, concerns from `HkRQ` about `attention as connectivity` remain unaddressed in a satisfying manner.

**Reviewer Scores:**

I think it is possible that `HkRQ` would raise their score, but I of course am not certain.

---

### Decision · Program_Chairs · 2026-01-26

Reject